# Learning Minimum-Size BDDs: Towards Efficient Exact Algorithms

**Christian Komusiewicz** [* 1]  **André Schidler** [* 2 3]  **Frank Sommer** [* 2]  **Manuel Sorge** [* 2]  **Luca Pascal Staus** [* 1]

## Abstract

Binary decision diagrams (BDDs) are widely applied tools to compactly represent labeled data as directed acyclic graphs; for efficiency and interpretability reasons small BDDs are preferred. Given labeled data, minimizing BDDs is NP-complete and thus recent research focused on the influence of parameters such as the solution size $s$ on the complexity [Ordyniak et al., AAAI 2024]. Our main positive result is an algorithm that is efficient if in particular $s$, the domain size $D$, and the Hamming distance between any two data points is small, improving on previous running-time bounds. This algorithm is inspired by the witness-tree paradigm that was recently successful for computing decision trees [Komusiewicz et al., ICML 2023], whose extension to BDDs was open. We extend our algorithmic results to the case where we allow a small number of misclassified data points and complement them with lower bounds that show that the running times are tight from multiple points of view. We show that our main algorithm holds practical promise by providing a proof-of-concept implementation.

## 1. Introduction

Binary decision diagrams (BDDs), also known as decision graphs and decision streams, are fundamental data structures used to describe (Lee, 1959; Akers, 1978; Bryant, 1992) and classify data (Oliver, 1993; Oliveira & Sangiovanni-Vincentelli, 1996; Mues et al., 2004; Shotton et al., 2013; Ignatov & Ignatov, 2017; Florio et al., 2023). BDDs have a wide variety of applications, ranging from circuit verification (Lee, 1959; Akers, 1978; Bryant, 1992) and optimization (see the survey by Castro et al. (2022)) to artificial intelligence topics such as planning (Sanner et al., 2010;

Castro et al., 2019), knowledge compilation (Abío et al., 2012; Lai et al., 2013; Serra, 2020) and constraint propagation (Andersen et al., 2007; Perez & Régin, 2015; Verhaeghe et al., 2018; Latour et al., 2019). In the context of classification, a BDD is a rooted directed acyclic graph in which each internal node represents a separating hyperplane and each terminal node is uniquely associated with a class. For reasons of efficiency of working with BDDs and interpretability, it is crucial that the size of the BDDs is small; throughout the paper by *size* we mean the number of inner nodes.

Compared to decision trees, another widely-studied concept with particular relevance to explainable AI, BDDs can offer smaller sizes and less redundancy (Oliver, 1993; Oliveira & Sangiovanni-Vincentelli, 1996; Shotton et al., 2013; Ignatov & Ignatov, 2017). Optimizing in particular the size of decision trees has received immense attention from the point of view of implementations (see the surveys by Carrizosa et al. (2021); Costa & Pedreira (2023)) and algorithms and complexity (Ordyniak & Szeider, 2021; Eiben et al., 2023; Komusiewicz et al., 2023; Ordyniak et al., 2024; Gahlawat & Zehavi, 2024; Schidler & Szeider, 2024; Harviainen et al., 2025; Kobourov et al., 2025). Computing minimum-size BDDs is comparatively less studied (Hu et al., 2022; Florio et al., 2023; Cabodi et al., 2024; Ordyniak et al., 2024). While both are NP-hard (Hyafil & Rivest, 1976; Bollig & Wegener, 1996), one reason for this difference may be that the search space of possible solutions is much larger than for decision trees (Florio et al., 2023); the topology needs to be optimized jointly with choosing the correct separating hyperplanes. Therefore it is even more important to find ways to shrink and efficiently traverse the search space. In this work we contribute to that research direction.

An algorithmic paradigm that was successful in the context of decision trees is that of witnesses (Komusiewicz et al., 2023): In this paradigm one builds the tree incrementally, identifying each time an incorrectly classified, *dirty*, example, which necessitates a new separating hyperplane that classifies the example correctly. Then one uses the dirty example as a witness that this separating hyperplane is necessary, mandating that the example is classified in the corresponding subtree. This reduces the search space and led to a new state of the art in computing minimum-size perfect decision trees, that is, decision trees with zero misclassifications (Staus et al., 2025).

---

[*]Equal contribution  [1]Institute of Computer Science, University of Jena, Germany [2]Institute of Logic and Computation, TU Wien, Austria [3]Computer Architecture, University of Freiburg, Germany. Correspondence to: Luca Pascal Staus <luca.staus@uni-jena.de>.

It was previously open if and how the witness paradigm could be applied to computing minimum-size BDDs (Ordyniak et al., 2024). Indeed, only impractical enumerative approaches were known with large running-time guarantees. In this paper, we show that the witness paradigm is directly applicable to computing BDDs, that it yields practically relevant algorithms with much improved running-time guarantees, and we give a proof-of-concept implementation that can already compute minimum-size BDDs of sizes up to 7.

**Problem statements.** In the learning problem for BDDs, we are given a set $E \subseteq \mathbb{R}^d$ of training data, labeled with classes by a labeling function $\lambda$. We may first aim to find the minimum-size BDD that perfectly classifies the training data, that is, there are no classification errors. This can be formulated as a search problem as follows:

> BOUNDED-SIZE BDD (BSBDD)
> *Instance:* A data set $(E, \lambda)$ and an integer $s \in \mathbb{N}$.
> *Task:* Find a BDD of size at most $s$ that (perfectly) classifies $(E, \lambda)$.

We call the corresponding size-minimization problem MINIMUM-SIZE BDD (MSBDD). Note that algorithms solving BSBDD can solve MSBDD with an additional factor of $s$ in the running time by successively increasing $s = 1, 2, \ldots$ and solving BSBDD until a solution is found.

The requirement of perfect classification is often too strong for practice and thus we also study the generalization ERROR BSBDD of BSBDD in which we are additionally given an integer $t$ and we ask to find a BDD of size at most $s$ that classifies all examples in $E$ correctly except for at most $t$ misclassifications.

Finally, researchers often study ordered BDDs (OBDDs) (Akers, 1978; Hu et al., 2022; Cabodi et al., 2024). An OBDD is associated with an order over the possible separating hyperplanes and each root-terminal path in the desired BDD has to respect this order. In the corresponding problem variants we replace BDD with OBDD in the problem name and ask for an OBDD instead.[1]

**Methods.** All of the above problem variants are NP-hard and thus we cannot expect efficient algorithms in general. Thus, we apply an approach based on parameterized algorithms (Gottlob et al., 2002; Flum & Grohe, 2006; Niedermeier, 2006; Cygan et al., 2015; Downey & Fellows, 2013): We try to identify small parameters $p$ of the input or desired output and design *fixed-parameter* (FPT) algorithms with running time $f(p) \cdot |I|^{O(1)}$, where $|I|$ is the input size. Or we try to prove that such algorithms are not possible, based on reasonable, widely accepted complexity-theoretic assumptions such as the Exponential Time Hypothesis (ETH) (Im-

pagliazzo & Paturi, 2001; Impagliazzo et al., 2001).

Relevant parameters for this analysis are the size $s$ of the desired (O)BDD, the number $d$ of features, the largest domain size $D$ of a feature, and the largest number $\delta$ in which two examples in $E$ differ[2]. To our knowledge, the only work performing such an analysis is Ordyniak et al. (2024) who obtained an FPT algorithm for BSBDD and its ensemble version with respect to $\delta$, $D$, and $s$.

**Results.** Our main result is an algorithm that applies the witness paradigm to BDDs in a nontrivial way: We show that a BDD can be built incrementally, by successively identifying an incorrectly classified, *dirty*, example $e \in E$, and finding a separating hyperplane that changes its current, incorrect classification path in the BDD. By doing that, we learn the new information that in all solutions the example $e$ needs to reach this separating hyperplane in its classification path. We can thus label a new edge in the BDD with $e$ as a witness and mandate that $e$ has to traverse this edge. This reduces the search space tremendously. The result is an algorithm for BSBDD with running time $\mathcal{O}((6s^2 \delta D)^s \cdot sn)$ (Theorem 3.3), where $n = |E|$, improving on the previously best-known $\mathcal{O}((3\delta D)^{s^2} \cdot n^{O(1)})$ (Ordyniak et al., 2024). Moreover, it avoids inherently enumerative approaches and instead is a search-tree algorithm with large potential for heuristic optimizations such as early termination rules. Our algorithmic approach is very versatile: We show that it can be extended to classification with errors (ERROR BSBDD), to computing OBDDs, and to computing minimum size ensembles of BDDs. Indeed, it shows that BDDs are strongly extendable, a desirable algorithmic property of data structures that represent data, which was previously open (Ordyniak et al., 2024). Apart from other algorithmic results, we provide several tightness results based on the ETH that show, for instance, that the dependency of the exponent on $s$ is optimal, and that dependency on other parameters of the exponential part of the running time cannot be removed. Proofs marked with (★) are deferred to the appendix.

To demonstrate the practical potential, we provide an implementation of our main algorithmic result. We compare it to a state-of-the-art SAT-based approach (Cabodi et al., 2024) on standard benchmark data. Despite its proof-of-concept status, our implementation computes optimal BDDs of size up to seven, is faster than the SAT-based approach for BDD sizes up to four, and in general solves 79% of the instances that also the SAT-approach solves within a 1h time limit.

## 2. Preliminaries

For $n \in \mathbb{N}$, we denote $[n] := \{1, 2, \ldots, n\}$ . For a vector $x \in \mathbb{R}^d$, we denote by $x[i]$ the $i$th entry in $x$. Let $\Sigma$

---

[1]Note that we do not assume that an ordering is given. The ordering has to be computed alongside the OBDD.

[2]See Ordyniak & Szeider (2021, Table 1) and Staus et al. (2025, Table 3) for indication that this parameter is small in practical data.

be a set of *class symbols*. A *data set* with classes $\Sigma$ is a tuple $(E, \lambda)$ of a set of *examples* $E \subseteq \mathbb{R}^d$ and their class labeling $\lambda \colon E \to \Sigma$. Note that this definition captures data sets with ordered features because such features can be mapped into $\mathbb{R}$. We assume that $(E, \lambda)$ does *not* contain two examples with identical coordinates but different class labels. For a fixed data set $(E, \lambda)$, we let $n := |E|$ denote the *number of examples* and $d$ the *dimension* of the data set.

For each dimension $i \in [d]$, we let $\mathsf{Thr}(i)$ be a smallest set of *thresholds* such that for each pair of examples $e_1, e_2 \in E$ with $e_1[i] < e_2[i]$, there is a threshold $h \in \mathsf{Thr}(i)$ with $e_1[i] \le h < e_2[i]$. Note that such a set $\mathsf{Thr}(i)$ can be computed in $\mathcal{O}(n \log n)$ time and that $|\mathsf{Thr}(i)| \le D$. A *cut* is a pair $(i, h)$ where $i \in [d]$ is a dimension and $h \in \mathsf{Thr}(i)$ is a threshold in dimension $d$. $\mathsf{Cuts}(E)$ is the set of all cuts. The *left side* of a cut with respect to $E' \subseteq E$ is $E'[\le (i, h)] := \{e \in E' : e[i] \le h\}$, and the *right side* of a cut with respect to $E'$ is $E'[> (i, h)] := \{e \in E' : e[i] > h\}$.

To define BDDs we use special directed acyclic graphs (DAGs). These DAGs $G$ have one root denoted by $\mathsf{root}(G)$ from which all other vertices are reachable. We write $V(G)$ to denote the set of vertices of $G$ and $A(G)$ to denote the set of arcs of $G$. The *arcs* in these DAGs are tuples $(s, h, c)$ where $s$ is the *start vertex*, $h$ is the *end vertex*, and $c$ is either $\ell$ or $r$ and indicates whether the arc goes to the *left child* or the *right child* of $s$. We say that $s$ is an *in-neighbor* of $h$ and $h$ is an *out-neighbor* of $s$. If only one arc with start vertex $s$ and end vertex $h$ exists or if the direction is irrelevant, then we just write $(s, h)$. We define the *in-degree* of a vertex $h$ as the number of arcs where $h$ is the end vertex and similarly we define the *out-degree* of a vertex $s$ as the number of arcs where $s$ is the start vertex. Additionally, each vertex is only allowed to have at most one left child and one right child, meaning the maximum out-degree is 2. A *path* in a DAG $G$ is a sequence of arcs $(a_1, \ldots, a_k)$ such that the end vertex of $a_i$ is the start vertex of $a_{i+1}$ for all $i \in [k-1]$. We write $\mathsf{Path}(G)$ to denote the set of all paths in $G$.

A *binary decision diagram (BDD)* is a special DAG $B$ as described above and each vertex in $V(B)$ is either a *split vertex* with out-degree 2 or a *leaf* with out-degree zero. Let $S(B)$ be the set of all split vertices and $L(B)$ the set of all leafs. Furthermore, $\mathsf{cut}_B : S(B) \to \mathsf{Cuts}(E)$ maps every split vertex to a cut and $\mathsf{cla}_B : L(B) \to \Sigma$ labels each leaf with a class. We drop the subscript if $B$ is clear. We assume that $\mathsf{cla}$ is injective meaning there can only be at most one leaf for each class in $\Sigma$. The *size* of $B$, denoted as $\mathsf{size}(B) = |S(B)|$, is the number of split vertices. Note that a BDD is not allowed to have two arcs with the same start and end vertex. Part a) of Figure 1 shows an example BDD with three split vertices $s_1, s_2, s_3$ and two leafs.

For each example $e$ we define its *classification path* $c_B(e) = (a_1, \ldots, a_\ell)$ in $B$ as the unique path from the root of $B$

to one of the leafs such that $a_i$ with start vertex $s_i$ goes to the left if and only if $e$ is on the left side of $\mathsf{cut}(s_i)$ for all $i \in [\ell]$. In other words, at each split vertex $s$ we decide if an example goes to the left or right child based on whether the example is on the left or right side of $\mathsf{cut}(s)$. For each vertex $v \in V(B)$ we define $E[B, v]$ as the set of all examples $e \in E$ where $v$ appears on $c_B(e)$. Similarly, for all arcs $a \in A(B)$ we define $E[B, a]$ as the set of all examples $e \in E$ where $a$ appears on $c_B(e)$. We say that these examples are *assigned* to $v$ and $a$, respectively. If $B$ is clear from the context, we just write $E[v]$. By definition, each example $e \in E$ is *assigned* to exactly one leaf $\ell$ of $B$. We say that $\ell$ is *the leaf of* $e$ in $B$ and denote $\ell$ by $\mathsf{leaf}(B, e)$ or just $\mathsf{leaf}(e)$ if $B$ is clear.

We call $B$ *reduced* if each arc has at least one example assigned to it. Clearly, any BDD can be turned into a reduced BDD without changing the leaf of any example by removing any vertex $v$ or arc $a$ with $E[v] = \emptyset$ or $E[a] = \emptyset$ and contracting any arc with an out-degree one start vertex.

An example $e \in E$ is *dirty* in $B$ if we have $\lambda(e) \ne \mathsf{cla}(\mathsf{leaf}(e))$. The set of all dirty examples in $B$ is $\mathsf{Dirty}(B)$. A BDD *classifies* $(E, \lambda)$ if no example is dirty. In this case we call $B$ *perfect*.

## 3. Witness-BDD-Algorithm

There is a naive brute force algorithm that solves BSBDD in polynomial time when $s$ is constant.

**Theorem 3.1 (★).** BOUNDED-SIZE BDD *can be solved in time* $(s^4 \cdot d \cdot D)^s \cdot |I|^{\mathcal{O}(1)}$.

### 3.1. An efficient Algorithm to solve BSBDD

Next, we present a much faster algorithm for BSBDD than Theorem 3.1 where we replace the number $d$ of dimensions with the much smaller parameter $\delta$. Our algorithm $\mathtt{WitBDD}$ is a branch-and-bound algorithm that starts with a BDD of size zero and then branches into all possible *refinements* of this BDD that increase its size by one. This is repeated recursively until the maximum size is reached. To limit the number of refinements that have to be considered, $\mathtt{WitBDD}$ annotates the arcs of the BDD with examples called *witnesses*. $\mathtt{WitBDD}$ must then ensure that if an arc is annotated with a witness, that witness must also be assigned to that arc. However, for this to work we first introduce a modified definition of BDDs called witness BDDs.

A *witness BDD (WBDD)* is a DAG $W$ as described in the previous section. Additionally, the root of $W$ must have in-degree zero and out-degree one. Each vertex in $V(W)$ that is not the root is either a *split vertex* with in-degree one and out-degree 2, a *merge vertex* with in-degree 2 and out-degree one or a *leaf* with in-degree one and out-degree zero. Let $S(W)$ be the set of all split vertices, $M(W)$ the set of

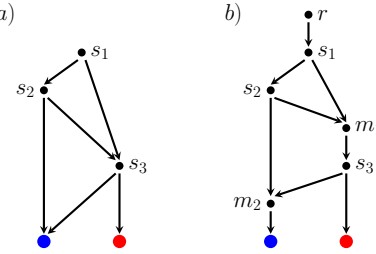

**Figure 1.** Part $a)$ shows a BDD $B$ and part $b)$ shows an equivalent WBDD $W$. We turn $B$ into $W$ by adding a new root vertex $r$ and the two merge vertices $m_1$ and $m_2$. To revert this transformation we contract the outgoing arcs of the three newly added vertices.

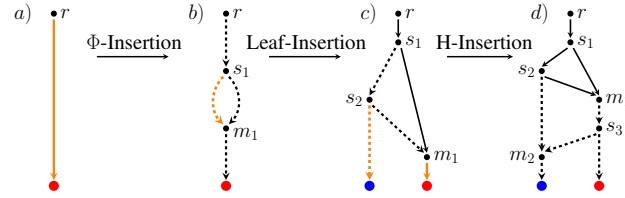

**Figure 2.** A sequence of one-step-refinements leading from a size-zero witness BDD to the witness BDD in part $b)$ of Figure 1. The arcs that get refined are drawn in orange and the resulting new arcs are dashed. First, a $\Phi$-Insertion creates the new split vertex $s_1$ and merge vertex $m_1$. Then, a Leaf-Insertion adds the new split vertex $s_2$ and new blue leaf. Finally, an H-Insertion adds another new split vertex $s_3$ and a new merge vertex $m_2$.

all merge vertices and $L(W)$ the set of all leafs. Furthermore, $\mathsf{cut}_W : S(W) \to \mathsf{Cuts}(E)$ maps every split vertex to a cut, $\mathsf{cla}_W : L(W) \to \Sigma$ labels each leaf with a class and $\mathsf{wit}_W : A(W) \to 2^E \setminus \{\emptyset\}$ assigns a non-empty set of examples called *witnesses* to each arc. We drop the subscript if $W$ is clear. We assume that $\mathsf{cla}$ is injective, meaning there can be at most one leaf for each class in $\Sigma$. The *size* of $W$, denoted as $\mathsf{size}(W)$, is the number of split vertices. Since we know the in- and out-degree of every vertex we obtain the equation

$$\mathsf{size}(W) = |S(W)| = |M(W)| + |L(W)| - 1 \quad (1)$$

which shows that the total vertex count is $2 \cdot \mathsf{size}(W) + 2$. Note that a WBDD is explicitly allowed to have two arcs with the same start and end vertex as long as one goes to the left and one goes to the right. We need this property only during branching in $\mathtt{WitBDD}$; the result will not have such arcs. Part b) of Figure 1 shows an example WBDD with three split vertices $s_1, s_2, s_3$, two merge vertices $m_1, m_2$, the root vertex $r$, and two leafs. All remaining definitions from the previous section are the same for WBDDs. We call a WBDD $W$ *consistent* if for all examples $e \in E$ the set of arcs $\{a \in A(W) : e \in \mathsf{wit}(a)\}$ forms a path that is a prefix of the classification path $c_W(e)$ of $e$ in $W$. In particular, this means we have $\mathsf{wit}(a) \subseteq E[W, a]$ for all $a \in A(W)$.

In the appendix we show that a perfect reduced BDD of size $s$ exists if and only if a perfect consistent WBDD of size $s$ exists. For an example see Figure 1. Thus, we only need to consider consistent WBDDs in $\mathtt{WitBDD}$.

We now define one-step-refinements for consistent WBDDs. Given a consistent WBDD $W$, a *one-step-refinement* modifies $W$ by adding a new split vertex and either a new leaf or a new merge vertex. Formally, we define a one-step-refinement as a tuple $(a_1, a_2, i, h, e)$ where $a_1$ and $a_2$ are arcs in $W$, $(i, h) \in \mathsf{Cuts}(E)$ is a cut and $e \in \mathsf{Dirty}(W)$ is a dirty example that is assigned to $a_1$. In particular this means that $a_1$ has to be on the classification path of $e$. Additionally, $a_2$ is allowed to be $\bot$.

There are three types of one-step-refinements which are demonstrated in Figure 2. For all three types, we first subdivide the arc $a_1$ to add a new split vertex $s$ with $\mathsf{cut}(s) = (i, h)$. The endpoint $q$ of the arc $a_1$ becomes the left child of $s$ if $e \in E[\leq (i, h)]$, and $q$ becomes the right child otherwise. Which vertex becomes the second child of $s$ depends on the type of the one-step-refinement.

The first type is called a *leaf-insertion*. A one-step-refinement is a leaf-insertion if there is no leaf $\ell$ with $\mathsf{cla}(\ell) = \lambda(e)$ and $a_2 = \bot$. For such a refinement we add a new leaf with this class and make it the second child of $s$.

The second type is called an *H-insertion*. A one-step-refinement is an H-insertion if it is not a leaf-insertion and $a_1 \neq a_2$. For such a refinement we subdivide the arc $a_2$ to add a new merge vertex $m$ which becomes the second child of $s$.

The third type is called a $\Phi$-*insertion*. A one-step-refinement is a $\Phi$-insertion if $a_1 = a_2$. Recall that we first subdivide arc $a_1$ to create a new split vertex $s$ and second, we subdivide the new arc $(s, q)$ to add a new merge vertex $m$ which becomes the second child of $s$.

When a one-step-refinement subdivides an arc $a$ it creates two new arcs $a'$ and $a''$. We set $\mathsf{wit}(a') = \mathsf{wit}(a'') = \mathsf{wit}(a)$. For the arc $a_s$ from $s$ to the second child of $s$ we set $\mathsf{wit}(a_s) = \{e\}$. Additionally, we add $e$ to the $\mathsf{wit}$ set of all arcs on the classification path of $e$ up to and including the arc from the start vertex of $a_1$ to $s$.

Figure 2 shows how applying these three types of one-step-refinements to a WBDD may look.

We write $W \xrightarrow{r} R$ to denote that the WBDD $R$ was created by applying the one-step-refinement $r$ to $W$. Note that some one-step-refinements may create an inconsistent WBDD or a graph which is not even a DAG: This can happen with a one-step-refinement $(a_1, a_2, i, h, e)$ if the examples in $\mathsf{wit}(a_1)$

---

**Algorithm 1:** `WitBDD`

**Input:** A consistent WBDD $W$, a data set $(E, \lambda)$, and a maximum size $s \in \mathbb{N}$.

**Output:** A perfect consistent WBDD of size at most $s$ or $\perp$ if none could be found.

1 **Function** `Refine` *($W$, $(E, \lambda)$, $s$)*
2     **if** $W$ *classifies* $(E, \lambda)$ **then return** $W$
3     **if** $W$ *has size* $s$ **then return** $\perp$
4     $e \leftarrow$ some dirty example from $\mathsf{Dirty}(W)$
5     **forall** $r = (a_1, a_2, i, t, e) \in Ref(W, e)$ **do**
6         Apply $r$ to $W$ to create a new consistent WBDD $R$
7         $R' \leftarrow$ `Refine` *($R$, $(E, \lambda)$, $s$)*
8         **if** $R' \neq \perp$ **then return** $R'$
9     **return** $\perp$

---

and the example $e$ are not on different sides of the cut $(i, h)$, if the new arc from $a_1$ to $a_2$ creates a cycle, or if $a_2 = \perp$ but a leaf with the class of $e$ already exists. No such refinement will lead to a perfect BDD. Thus, we write $\mathsf{Ref}(W)$ to denote the set of all *valid* one-step-refinements, that is, all one-step-refinements where $R$ is still a consistent WBDD. We write $\mathsf{Ref}(W, e)$ to denote the subset of valid one-step-refinements where $e$ is the dirty example.

We can now describe the branching algorithm `WitBDD`. Initially we choose any example $e \in E$ and construct a consistent WBDD $W$ with exactly one arc between the root $r$ and a leaf $\ell$. We set $\mathsf{wit}((r, \ell)) = \{e\}$ and $\mathsf{cla}(\ell) = \lambda(e)$. We then call `Refine` in Algorithm 1. `Refine` takes a consistent WBDD $W$ as input and checks if a sequence of one-step-refinements exists that can turn $W$ into a perfect consistent WBDD with size at most $s$. For this, we first check if $W$ is already perfect. If not, we then check if $W$ has already reached the maximum size. If that is also not the case we choose some dirty example $e$ in Line 4. We then iterate over all possible valid one-step-refinements with $e$ as the dirty example in Line 5, apply them to $W$ in Line 6 and recursively call `Refine` in Line 7. If a perfect consistent WBDD is found that has size at most $s$ we immediately return it. Otherwise the loop runs until all possible one-step-refinements have been checked.

**Lemma 3.2.** *For a given data set $(E, \lambda)$ and positive integer $s$,* `WitBDD` *correctly solves* BSBDD.

Before we prove Lemma 3.2, we will first show that the algorithm has the desired running time.

**Theorem 3.3.** BSBDD *can be solved in* $\mathcal{O}((6s^2\delta D)^s \cdot sn)$ *time.*

*Proof.* We first bound how many times `Refine` is called. Clearly, the recursion depth is at most $s$ since the size of

the WBDD is bounded by $s$. That means that `Refine` is called $\mathcal{O}(R^s)$ times where $R$ is the maximum number of iterations of the loop in Line 5. Since the loop is not called if $W$ already has size $s$ we can assume that we have $\mathsf{size}(W) < s$.

For a one-step-refinement in $\mathsf{Ref}(W, e)$ we first need to choose an arc $a_1$. This arc must be on the classification path of $e$. This path can visit each non-leaf vertex at most once and contains exactly one leaf. Due to Equation (1) we know that there are at most $2 \cdot \mathsf{size}(W) + 2 \leq 2s$ vertices in $W$. Hence, the number of arcs on the classification path of $e$ is also at most $2s$. For $a_2$ we can choose any arc in $A(W)$. To bound this, we can add up the out-degrees of all vertices. This gives us $2 \cdot |S(W)| + |M(W)| + 1$. Since we have at least one leaf and $\mathsf{size}(W) < s$ we get the following inequality:

$$2 \cdot |S(W)| + |M(W)| + 1 \leq 3 \cdot \mathsf{size}(W) + 1 \leq 3s.$$

Hence, there are at most $3s$ choices for $a_2$. Since the witnesses of $a_1$ and $e$ must be on different sides of the cut $(i, h)$ we have at most $\delta$ choices for the dimension $i$ and at most $D$ choices for the threshold $h$.

This bounds the total number of one-step-refinements that are considered in Line 5 by $6s^2\delta D$. Applying a one-step-refinement can be done in $\mathcal{O}(sn)$ time and checking if all examples are classified correctly and if not finding a dirty example can also be done in $\mathcal{O}(sn)$ time. $\quad\square$

To prove Lemma 3.2 we first need to define embeddings. An embedding maps each arc in a WBDD $W$ to a path in a different WBDD $W'$ subject to certain conditions.

**Definition 3.4.** Given two consistent WBDDs $W$ and $W'$ we say that $W$ *admits an embedding* $\varphi : A(W) \rightarrow \mathsf{Path}(W')$ into $W'$ if the following conditions hold:

1. $S(W) \subseteq S(W')$ and $M(W) \subseteq M(W')$ and $L(W) \subseteq L(W')$ and $\mathsf{root}(W) = \mathsf{root}(W')$.

2. For all $s \in S(W)$ we have $\mathsf{cut}_W(s) = \mathsf{cut}_{W'}(s)$ and for all $\ell \in L(W)$ we have $\mathsf{cla}_W(\ell) = \mathsf{cla}_{W'}(\ell)$.

3. For all $a \in A(W)$ we have:

    (a) $\varphi(a)$ is a path from the start vertex of $a$ to the end vertex of $a$ and they are the only vertices from $V(W)$ that appear on $\varphi(a)$.

    (b) For all arcs $a'$ on $\varphi(a)$ we have $\mathsf{wit}_W(a) \subseteq \mathsf{wit}_{W'}(a')$.

    (c) If $s_1 \in S(W)$, then $a$ goes to the left in $W$ if and only if the first arc in $\varphi(a)$ goes to the left in $W'$.

4. For all $a_1, a_2 \in A(W)$ with $a_1 \neq a_2$, the only vertices appearing on both $\varphi(a_1)$ and $\varphi(a_2)$ may be their start and end vertices.

The idea of the proof is to first show that the starting WBDD of the algorithm admits an embedding into some perfect WBDD $P$ of size at most $s$ if such a WBDD exists. We then show that for any WBDD $W$ that admits an embedding into $P$ and that has at least one dirty example $e$ there exists a one-step-refinement that the algorithm considers that creates a WBDD that also admits an embedding into $P$. We then finish the proof by showing that if $\mathsf{size}(W) = \mathsf{size}(P)$, then $W$ must be perfect because $P$ is perfect.

*Proof of Lemma 3.2.* We assume that a perfect WBDD $P$ of size at most $s$ exists. If not then the algorithm will clearly return $\bot$ since any WBDD generated by the algorithm has size at most $s$. We assume that the witness labeling of $P$ is maximal, that is, $\mathsf{wit}_P(a) = E[P, a]$ for all $a \in A(P)$.

We start by showing that the WBDD $W$ of size zero that Refine is initially called with admits an embedding into $P$. We know that $W$ only has a root $r$ and one leaf $\ell$. The root of a WBDD is always the same and since $P$ is perfect, there must be a leaf in $P$ that has the same class as $\ell$. Hence, Items 1 and 2 of Definition 3.4 are fulfilled. We now just need to map the arc $(r, \ell)$ of $W$ to a path from $r$ to $\ell$ in $P$ such that Items 3a to 3c and 4 are fulfilled. Let $e$ be the witness in $\mathsf{wit}_W(r, \ell)$. Since $P$ is perfect, we know that the classification path of $e$ in $P$ must start in $r$ and end in $\ell$. Thus, by mapping $(r, \ell)$ to this path we fulfill all conditions.

Next, we assume that we already have some WBDD $W$ that admits an embedding $\varphi$ into $P$. We want to show that if Refine is called with $W$, then at least one of the WBDDs $R$ that are created inside the for-loop must also admit an embedding into $P$. Let $e$ be a dirty example in $W$. If such an example does not exist then $W$ is already perfect and we are done. We now need to find a one-step-refinement $r = (a_1, a_2, i, h, e) \in \mathsf{Ref}(W, e)$ with $W \xrightarrow{r} R$ such that $R$ admits an embedding $\tau$ into $P$.

We start by identifying $a_1$ and the vertex $z_1$ which is created when the one-step-refinement subdivides $a_1$. Let $a_1$ be the first arc on the classification path $c_W(e)$ of $e$ in $W$ such that $\varphi(a_1)$ is not fully included in $c_P(e)$ and let $z_1$ be the start vertex of the first arc in $\varphi(a_1)$ where the two paths do not match. For this to happen there must be exactly two arcs in $P$ with $z_1$ as the start vertex which means that $z_1$ is a split vertex. Note that such an arc $a_1$ must exist since $e$ is dirty in $W$ but not dirty in $P$. Let $s_1$ be the start vertex and $h_1$ the end vertex of $a_1$. We need to show that $z_1$ is not equal to $s_1$ or $h_1$ and therefore not a vertex in $W$.

Clearly, we have $z_1 \neq h_1$ since $h_1$ is not the start vertex of any arc in $\varphi(a_1)$. We also have $z_1 \neq s_1$. To see this we use Item 3c in Definition 3.4. Without loss of generality, we assume that $a_1$ goes to the left in $W$, that is, $a_1 = (s_1, h_1, \ell)$. Hence, $e$ is on the left side of $\mathsf{cut}_W(s_1)$. The first arc in $\varphi(a_1)$ must therefore also go to the left in $P$. By the

definition of $z_1$, the arc on $c_P(e)$ with start vertex $z_1$ must go to the right which is a contradiction to $e$ being on the left side of $\mathsf{cut}_W(s_1) = \mathsf{cut}_P(s_1)$. Hence, we have $z_1 \neq s_1$.

With this we now know that $z_1 \notin V(W)$. We now set $(i, h) = \mathsf{cut}_P(z_1)$. Without loss of generality, we assume that $e$ is on the right side of $(i, h)$. Due to Item 3b in Definition 3.4 and the definition of $z_1$ we know that all witnesses in $\mathsf{wit}(a_1)$ must be on the left side of $(i, h)$ while $e$ is on the right side. This ensures that the one-step-refinement $r$ creates a consistent WBDD and is therefore contained in $\mathsf{Ref}(W, e)$.

We now just need to find a suitable arc $a_2$. To do this we will first try to find the vertex $z_2$ which is created when the one-step-refinement subdivides $a_2$. This vertex must come after $z_1$ on the classification path of $e$ in $P$. Let $Q$ be the suffix of $c_P(e)$ that starts at the vertex following $z_1$. Also, let $\Psi$ be the image of $\varphi$, i.e. $\Psi = \{\varphi(a) : a \in A(W)\}$. We now define $z_2$ as the first vertex on $Q$ that also appears on some path $\pi \in \Psi$.

We now differentiate between two cases based on whether such a $z_2$ exists or not. We first assume that it does not exist. In this case there cannot be a leaf in $W$ that has the same class as $e$. If this leaf $\ell$ did exist it would have an in-neighbor $\ell'$ in $W$. The path $\varphi(\ell', \ell)$ would then intersect $Q$ which contradicts $z_2$ not existing. In this case we set $a_2 = \bot$ and the one-step-refinement becomes a leaf-insertion.

We now look at the other case where $z_2$ does exist. Let $a_2$ be the arc in $W$ with $\varphi(a_2) = \pi$. Also let $s_2$ be the start vertex and $h_2$ the end vertex of $a_2$. We need to show that $z_2$ is a merge vertex in $Z$ and that $z_2$ does not exist in $W$.

Let $a_2$ be the arc in $W$ with $\varphi(a_2) = \pi$. Also let $s_2$ be the start vertex and $h_2$ the end vertex of $a_2$. We first want to show that $z_2$ must be a merge vertex. To do this we assume towards a contradiction that $z_2$ only has one incoming arc. We now want to find an arc $a_1$ in $W$ such that $z_2$ is not the start vertex of $a'$ but still appears on $\varphi(a')$. If $z_2 \neq s_2$ we can just use $a' := a_2$. If $z_2 = s_2$ then we set $a'$ to the incoming arc of $z_2$ in $W$. Now, if $z_2$ is not the start vertex of $Q$ then the parent of $z_2$ in $P$ must appear on both $\varphi(a')$ and $Q$. This is a contradiction since $z_2$ should by definition be the first such vertex. If $z_2$ is the start vertex of $Q$ then $z_1$ must appear on the path $\varphi(a')$ in $P$. Due to condition Item 4 in Definition 3.4, we must have $a_1 = a'$. We know that $z_2$ appears on $c_P(e)$ right after $z_1$. We also know that $\varphi(a_1)$ diverges from that path at $z_1$ but still has $z_2 = h_1$ as its end vertex. That is only possible if $z_2$ has two incoming arcs which contradicts it having only one. This proves that $z_2$ must be a merge vertex.

We now want to show that $z_2$ is not equal to $s_2$ or $h_2$ and therefore not a vertex in $W$. For this let us assume towards a contradiction that $z_2$ is equal to $s_2$ or $h_2$. Since $z_2$ is a

merge vertex it has exactly two incoming arcs $a'$ and $a''$ in $W$ with start vertices $s'$ and $s''$. Each start vertex of the two incoming arcs of $z_2$ in $P$ must appear on at least on of the paths $\varphi(a')$ and $\varphi(a'')$. If $z_2$ is not the first vertex on $Q$ then one of those start vertices must also appear on $Q$ which contradicts $z_2$ being the first vertex on $Q$ that appears on some path in $\Psi$. If $z_2$ is the first vertex on $Q$ then $z_1$ must be one of those two start vertices. This means $z_2$ appears on exactly one of the paths $\varphi(a')$ and $\varphi(a'')$; $z_2$ cannot appear on both since that would violate Item 4 of Definition 3.4. Without loss of generality, we assume that $z_2$ appears on $\varphi(a')$ which means we must have $a' = a_1$. We know that the classification path $c_P(e)$ diverges from $\varphi(a_1)$ at $z_1$. But since $z_2$ follows $z_1$ on that path we know that $z_1$ must also appear on $\varphi(a'')$, contradicting our assumption that it appears on exactly one of the incoming paths.

We now know that $z_2 \notin V(W)$ and that it must be a merge vertex in $P$. With this we have found the two vertices that are created by the one-step-refinement when subdividing $a_1$ and $a_2$. If $a_1 = a_2$ then $r$ becomes a $\Phi$-insertion. Otherwise, it becomes an H-insertion.

Now, that we have found a suitable one-step-refinement $r$, we need to show that $R$ with $W \xrightarrow{r} R$ admits an embedding $\tau$ into $P$. Due to how $r$ was constructed we know that Items 1 and 2 of Definition 3.4 are fulfilled. We just need to deal with Item 3 of Definition 3.4. For this we define $\tau(a) = \varphi(a)$ for all arcs $a$ that exist in both $W$ and $R$. We now need to find a suitable path in $P$ for all arcs that are added by $r$. The arcs that are created by subdividing $a_1$ or $a_2$ are assigned to the corresponding subpaths of $\varphi(a_1)$ and $\varphi(a_2)$. By construction the newly added vertices $z_1$ and $z_2$ must exist on these paths. Next we need to assign a path to the arc $a_z$ from $z_1$ to $z_2$. For simplicity we set $z_2$ to the newly added leaf in the case that $r$ is a leaf insertion. We define $\tau(a_z)$ as the subpath of $c_P(e)$ that starts at $z_1$ and ends at $z_2$. By definition of $z_1$ and $z_2$, this path must exist and $z_1$ and $z_2$ must be the only vertices from $V(R)$ that appear on it. With this, Item 3a is fulfilled. Item 3c of Definition 3.4 is also fulfilled by construction. For all arcs $a$ on the classification path of $e$ in $R$ up to $z_2$ the path $\varphi(a)$ must be fully included in the classification path of $e$ in $P$. Item 3b of Definition 3.4 must also be fulfilled since these are the only arcs where $e$ is added as a witness and when subdividing an arc the witnesses are just copied from the original arc. Since we chose $z_1$ and $z_2$ as the first vertices that fulfill their respective conditions we clearly have that no two paths overlap in $P$ outside of their start and end vertices. Hence, Item 4 is also fulfilled. This proves the statement.

Finally, to prove the correctness of the algorithm we only need to show that $W$ is perfect if $\mathsf{size}(W) = \mathsf{size}(P)$. From Item 1 of Definition 3.4 and Equation (1) we see that we must have $V(W) = V(P)$. This also means that any

arc $a$ in $W$ is mapped to itself by $\varphi$. With Item 3c this guarantees that the structure of $W$ must be the same as the structure of $P$. Finally, with Item 2 we can see that the classification paths of all examples will be the same in $W$ and $P$. Consequently, $W$ must also be perfect. $\qquad\square$

## 3.2. Extensions of `WitBDD` for more general Problems

Ordyniak et al. (2024) presented a general theoretical framework for learning smallest interpretable models, such as decision trees and BDDs. More precisely, they presented a framework of so-called strong-extendability to compute perfect decision trees and a weaker framework of so-called extendability to compute perfect BDDs. With the help of extendability, they showed that BSBDD can be solved in $(\delta \cdot D \cdot 2^{\mathcal{O}(s)})^s$ time. They posed as an open question whether this running time can be significantly reduced.

Next, we argue that BDDs are in fact strong-extendable by invoking `WitBDD`. In order to verify this, an annotated model $(M, A)$ is required such that $M$ is not a model for the input classification instance and $A$ is some annotation. Moreover, let $e$ be a dirty example, that is, an example which is not correctly classified by $M$. A *full set of strict extensions for an annotated model* $(M, A)$ and example $e$ is a set $\mathcal{E}$ of strict extensions of $(M, A)$ such that every model $M'$ that correctly classifies $e$ and is an extension of $(M, A)$ also has an extension of some annotated model in $\mathcal{E}$. Now, observe that our notion of a witness BDD together with the definition of one-step-refinements and embeddings, and the correctness of our algorithm `WitBDD` shown in Lemma 3.2 verifies that BDDs are strongly extendable.

Ordyniak et al. (2024, Theorem 3) showed that strong-extendability leads to an efficient algorithm to solve *ensembles*, that is, a set $\mathcal{S}$ of BDDs with at least one split vertex each, where the classification of an example is the majority vote of all BDDs in $\mathcal{S}$. By $s$ we denote the total size of all BDDs in the ensemble. We denote the ensemble variant of a problem by adding ENSEMBLE to the problem name. Thus, we obtain the following.

**Corollary 3.5.** ENSEMBLE BSBDD *can be solved in time* $\mathcal{O}((6s^2\delta D)^s \cdot sn)$.

Moreover, the framework of strong-extendability is not only limited to perfect BDDs; it can also be applied to ERROR BSBDD, that is, computing a minimum size BDD with at most $t$ errors: Now it is not sufficient anymore to choose an arbitrary dirty example in Line 4 of Algorithm 1. If there are at most $t$ errors, and the WBDD has size at most $s$, we found a solution. Otherwise, we choose an arbitrary subset $S$ of $t + 1$ dirty examples. If a solution exists, at least one of them has to be classified correctly. Next, we branch on all one-step-refinements for each dirty example in $S$ in Line 5. The correctness of this algorithm can be shown analogously

to Lemma 3.2 and thus we obtain the following:

**Corollary 3.6.** ERROR BSBDD *and* ENSEMBLE ERROR BSBDD *can be solved in time* $\mathcal{O}((6s^2\delta D(t+1))^s \cdot sn)$.

Moreover, as we discussed in the introduction, WitBDD can not only be used to solve decision problem BSBDD (and its generalization), but WitBDD can also be used to solve the optimization variant MSBDD. Thus, to solve the optimization problems, we obtain an additional factor of $s$ in the running times.

Also, WitBDD works for OBDDs by simply checking before Line 7 if $R$ is ordered and only calling Refine if that is the case. Hence, we obtain the following.

**Corollary 3.7.** BOUNDED-SIZE OBDD *can be solved in time* $\mathcal{O}((6s^2\delta D)^s \cdot s^2n)$.

Note that this generalizes to the ensemble model and the model with up to $t$ errors.

## 4. Further Parameterized Complexity Results

Next, we study whether ERROR BSBDD can be solved efficiently when the number $d$ of dimensions is small. Kobourov et al. (2025, Theorem 1) showed that minimum-size decision trees with zero errors, that is, $t = 0$, can be computed in polynomial time when $d$ is a constant. In the conclusion they argue that also a minimum-size decision tree with at most $t$ errors can be computed in polynomial time whenever $d$ is a constant. We now show that, in contrast, ERROR BSBDD remains NP-hard even if $d = 4$.

**Theorem 4.1 (★).** ERROR BSBDD *and* ERROR BSOBDD *are NP-hard and cannot be solved in* $f(s + d) \cdot n^{o(s+d)}$ *time, unless the ETH fails, even if* $d = 4$.

In contrast, when the number of dimensions $d$ and the maximum domain size $D$ are bounded, then minimum-size BDDs can be computed efficiently also for unbounded error parameter $t$. This result also holds for OBDDs.

**Theorem 4.2 (★).** ERROR BSBDD *and* ERROR BSOBDD *are FPT for* $D + d$.

Ordyniak & Szeider (2021) showed that computing a minimal size decision tree is unlikely to admit an FPT algorithm for $s$, even if $D = 2$ and $t = 0$ and that it is NP-hard even if $D = 2$, $\delta = 2$, and $t = 0$. Their reduction works for BDDs and OBDDs as well giving the following hardness.

**Proposition 4.3 (★).** BSBDD *and* BSOBDD *(1) cannot be solved in* $f(s + D) \cdot n^{o(s+D)}$ *time, unless the ETH fails, even if* $D = 2$, *and (2) are NP-hard, even if* $D = \delta = 2$.

Recall that our witness-BDD algorithm presented in Theorem 3.3 has a running time of $\mathcal{O}((6s^2\delta D)^s \cdot sn)$. Proposition 4.3 shows that the exponential dependence on $s$ cannot be significantly improved without violating the ETH.

## 5. Experiments

We implemented Algorithm 1 to gauge how it performs in practice. More precisely, our implementation solves MS-BDD and MSOBDD, the optimization problems where we search for a perfect (O)BDD of minimal size. We are not aware of existing implementations for MSBDD and hence compare against the SOTA SAT-implementation for MSOBDD due to Cabodi et al. (2024). Since the implementation of Cabodi et al. was not publicly available, we re-implemented their SAT-encoding.

**Experimental setup.** For our experiments we used 35 datasets of the Penn Machine Learning Benchmarks (Romano et al., 2022) that were also used by Staus et al. (2025) in their experiments on computing optimal decision trees; see Table 1 in the appendix. Each of these instances is a binary classification problem. Analogously to Cabodi et al. (2024), we randomly sampled multiple subsets of the examples from each data set: Specifically, for each data set we constructed 10 instances by randomly selecting $10\%$ of the examples and 10 instances by randomly selecting $20\%$ of the examples. This gives a total of 700 instances. For each instance we set a time limit of 60 minutes. Our experiments were performed on Intel(R) Xeon(R) Platinum 8360Y(2) CPUs with 2.6 GHz and 24 cores and 256 GB RAM. Each individual experiment was performed on a private core with a RAM limit of 2 GB. For the SAT encoding we increased the limit to 20 GB. We implemented WitBDD in C++ and the SAT-formulation in Python with CaDiCaL 1.9.5 as the SAT-solver. Our source code is available in the supplementary material.[3]

**Heuristic running time improvements.** Compared to the pure theoretical description, WitBDD incorporates some first heuristic improvements. Some improvements are also used for the SAT-encodings to make the comparison fair.

1) Data reduction rules. For WitBDD and for the SAT-encoding, we apply some simple data reduction rules that were described by Staus et al. (2025) for computing minimum-size decision trees. These rules reduce redundancy in the dataset by removing examples that have the same value as another example in all dimension, dimension where all examples have the same value, and cuts that are equivalent to other cuts. It is straightforward to show that these rules can also be applied to BSBDD For further details, we refer to Staus et al. (2025, Appendix B).

2) Dirty example priority. Any dirty example $e$ in Line 4 of Algorithm 1 is possible. Ideally, we want $e$ to minimize the number of recursive calls in Line 5. Since it takes too long to calculate this number exactly for all dirty examples $e$, we instead try to approximate it as follows for each $e$: We count

---

[3]https://doi.org/10.5281/zenodo.15489411

the number of cuts that separate $e$ and the witness of the last arc on the classification path of $e$ and multiply this number by the length of the current classification path $e$.

3) Budget one improvement. If the difference of the size of the current WBDD and the maximum size $s$ is only one then all dirty examples must be correctly classified with a single one-step-refinement $\rho$. Thus, when `WitBDD` chooses an arc $a_1$ for $\rho$, all dirty examples must be assigned to $a_1$; otherwise we cannot classify them all correctly. Consequently, we can discard all arcs $a_1$ which violate this property.

4) Pair lower bound. Staus et al. (2025, Section 4) describe a Pair Lower Bound (PairLB) for computing the smallest perfect decision tree. This bound is also valid for BDDs: In the PairLB one constructs a SET COVER instance where the universe consists of all pairs of examples of different classes. For each cut $c$ in the dataset, a subset consisting of all pairs that are separated by $c$ is added to the SET COVER instance $I$. The set $X$ of all cuts in a perfect BDD is a solution to $I$. Hence, any lower bound for $I$ is a lower bound for the smallest size of any perfect BDD. To compute such a lower bound, we solve the LP relaxation of an ILP formulation for SET COVER, using Gurobi 10.0.3. (Gurobi Optimization, LLC, 2024). This lower bound is used as starting value for $s$ in `WitBDD` and the SAT-encoding.

5) Packing lower bound. For `WitBDD`, we additionally use the SET COVER instance $I$ from the PairLB to compute a packing of the cuts such that each set in the packing has at least one example pair that can only be separated by cuts in that set. The number of sets in the packing is a lower bound for $I$ and therefore also for the smallest size of a perfect BDD. The advantage is that this lower bound can also be used during the run-time of `WitBDD`: At the beginning of each recursive call, we can check how many sets of packing are not yet covered by some cut in the current BDD. This number is a lower bound for the number of further one-step-refinements needed to correctly classify all examples.

For `WitBDD` we simultaneously use five different packings to improve the likelihood that in some packing the number of uncovered sets is too large. The first packing is computed greedily while the remaining four are computed randomly.

**Experimental results.** `WitBDD` solved $443$ instances and the SAT encoding solved $557$ out of the $700$ instances. All instances that were solved by `WitBDD` were also solved by the SAT encoding. Figure 3 shows the running times of the two algorithms compared to the sizes of the optimal BDDs. It shows that `WitBDD` generally performs better than the SAT encoding on instances with a small optimal size. In fact, on the $353$ instances with an optimal size of at most $4$, `WitBDD` is on average roughly $828$ times faster than the SAT encoding with a median speedup of $64$. However, on the remaining $90$ instances with an optimal size that is

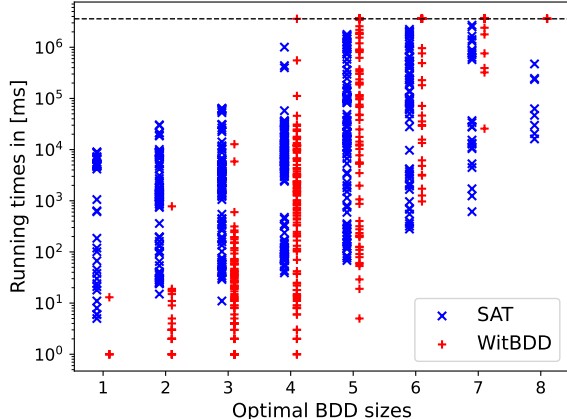

*Figure 3.* Comparison of `WitBDD` and the SAT encoding based on optimal BDD size. The dashed line represents the timeout.

bigger than $4$ the SAT encoding was on average roughly $37$ times faster than `WitBDD` with a median speedup of $7$.

The reason for this is most likely that the search space of `WitBDD` increases by a lot as the size bound $s$ increases. For instances with an optimal size of at least $5$ the function `Refine` was called on average roughly $360 \cdot 10^6$ times with a median of $14 \cdot 10^6$. On instances with an optimal size $< 4$ the average was only $4 \cdot 10^6$ with a median of $3038$.

In a nutshell, the experiments show that our basic proof-of-concept implementation outperforms the SOTA for small BDD sizes ($s \leq 4$) while it is substantially slower for medium-size BDDs ($5 \leq s \leq 8$) and larger BDDs ($s > 9$) are out of reach of the current methods.

## 6. Conclusion

We provided a branch-and-bound algorithm `WitBDD` with running time $\mathcal{O}((6s^2\delta D)^s \cdot sn)$ for computing minimal size BDDs. This significantly improves upon the previous best known running time guarantee of $(\delta \cdot D \cdot 2^{\mathcal{O}(s)})^s$ (Ordyniak et al., 2024). Moreover, we provided a proof-of-concept implementation `WitBDD` and showed that `WitBDD` holds practical promise against the SOTA SAT-formulation. In our opinion, `WitBDD` holds a lot of potential for further improvements: Since it is a branch-and-bound algorithm, symmetry breaking techniques and improved lower bounds should give a substantial speed-up.

While we presented our results to compute minimal-*size* BDDs, our techniques also work for computing minimal-*depth* BDDs: in Algorithm 1 we just need to restrict the one-step-refinements to those that do not violate the depth constraint. Since each BDD with depth at most $h$ has at most $2^h$ inner nodes, this gives an $\mathcal{O}((6h^2\delta D)^{2^h} \cdot 2^h n)$ time algorithm for this problem.

## Acknowledgments

André Schidler was supported by Austrian Science Fund (FWF) grant 10.55776/P36420 and an Amazon Research Award (Fall/2023). Frank Sommer was supported by the Alexander von Humboldt Foundation. Luca Pascal Staus was supported by the Carl Zeiss Foundation, Germany, within the project "Interactive Inference".

## Impact Statement

This paper presents work whose goal is to advance the field of Machine Learning. There are many potential societal consequences of our work, none which we feel must be specifically highlighted here.

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

# Appendix

## A. Additional Material for Section 3

### A.1. Proof of Theorem 3.1

*Proof.* Let $(I, s)$ be an instance of BSBDD. We can assume $|\Sigma| \leq s + 1$ since a BDD can have at most $s + 1$ leafs and therefore cannot correctly classify all examples if they have more than $s + 1$ different classes.

The idea of the algorithm is to simply enumerate all possible BDDs of size $s$ and check if they are perfect. We start by creating $s$ split vertices and one designated leaf for each class in $\Sigma$ for a total of at most $2s + 1$ vertices. Each split vertex now needs a left child and a right child. There are at most $2s$ options for the endpoint of each arc. This means we can enumerate all possible BDD structures in $s^{4s} \cdot |I|^{\mathcal{O}(1)}$ time.

For each BDD structure we now need to go through all possible cuts for each split vertex. Since there are $d$ dimensions and at most $D$ thresholds in each dimension, this step requires $d^s \cdot D^s \cdot |I|^{\mathcal{O}(1)}$ time for each BDD structure.

Checking if a BDD is perfect can be done in $|I|^{\mathcal{O}(1)}$ time. Hence, the overall running time is $(s^4 \cdot d \cdot D)^s \cdot |I|^{\mathcal{O}(1)}$. $\quad\square$

**Transforming a BDD into an WBDD and vice versa.** Given a reduced BDD $B$ we can turn $B$ into an equivalent consistent WBDD $B'$ with $\mathsf{size}(B) = \mathsf{size}(B')$ and $\mathsf{cla}_B(\mathsf{leaf}(B, e)) = \mathsf{cla}_{B'}(\mathsf{leaf}(B', e))$ for all $e \in E$; for an example see Figure 1. We first add a new root with exactly one arc going to the previous root $\mathsf{root}(B)$. Next, we need to make sure that all vertices in $B$ have in-degree one since they are all either split vertices or leafs. For a vertex $v \in V(B)$ with an in-degree of at least two we add a new merge vertex $m$ and replace the end vertex in two of the incoming arcs of $v$ with $m$. We also create a new arc $(m, v)$. We repeat this until all vertices $v \in V(B)$ have in-degree one except for the new root and the merge vertices. Finally, we set $\mathsf{wit}_{B'}(a) = E[B', a]$ for all $a \in A(B')$. Since $B$ is reduced, we know that these sets are not empty.

Similarly, we can turn any consistent WBDD $W$ into an equivalent reduced BDD $W'$ with $\mathsf{size}(W) = \mathsf{size}(W')$ and $\mathsf{cla}_W(\mathsf{leaf}(W, e)) = \mathsf{cla}_{W'}(\mathsf{leaf}(W', e))$ for all $e \in E$, see Figure 1. We just reverse the process described above by contracting the outgoing arcs of any merge vertex and the root. Hence, a perfect reduced BDD of size $s$ exists if and only if a perfect consistent WBDD of size $s$ exists. Thus, we only need to consider consistent WBDDs in `WitBDD`.

## B. Additional Material for Section 4

### B.1. Proof of Theorem 4.1

*Proof.* We provide a reduction from MULTICOLORED $k$-BICLIQUE, which cannot be solved in $f(k) \cdot |V(G)|^{o(k)}$ time unless the ETH fails (Cygan et al., 2015).

> MULTICOLORED $k$-BICLIQUE
> *Input:* An integer $k$ and a bipartite graph $G = (P, Q, E)$ where $P = \{P_1, \ldots, P_k\}$, $Q = \{Q_1, \ldots, Q_k\}$, and $|P_i| = n = |Q_i|$ for each $i \in [n]$.
> *Question:* Does $G$ have a multicolored $k$-biclique, that is, a vertex set $S$ such that $|S \cap P_i| = 1 = |S \cap Q_i|$ such that each vertex in $S \cap P$ is adjacent to each vertex in $S \cap Q$?

Note that it is safe to assume that $k \leq n$. The property that all partite sets $P_i$ and $Q_i$ have the same number $n$ of vertices is only used to simplify the proof. For each partite set $P_i$ and $Q_i$ we let $p_i^1, p_i^2, \ldots, p_i^n$ and $q_i^1, q_i^2, \ldots, q_i^n$ be an arbitrary but fixed ordering of $P_i$ and $Q_i$, respectively.

**Intuition:** We create two dimensions $d_P^\ell$ and $d_P^r$, and $d_Q^\ell$ and $d_Q^r$ per partite set $P$ and $Q$, respectively. Dimension $d_P^\ell$ is partitioned into $k$ different *blocks* $T_{P,\ell}^1, T_{P,\ell}^2, \ldots, T_{P,\ell}^k$ such that the $i$th block represents the vertex set $P_i$. The other dimensions are partitioned similarly.

In our construction we use many *badges*, that is, sets of indistinguishable examples. Each block starts with a badge of blue *forcing examples* of size more than $t$ and ends with a badge of red *enforcing examples* of size more than $t$ (or vice versa). Consequently, each minimal-size BDD (a) has to separate each two consecutive blocks with a cut to separate the red enforcing examples of the previous block and the blue forcing examples of the new block and (b) needs another cut within each block to separate the blue *forcing examples* and the red *enforcing examples* of that block to fulfill the error threshold $t$. We set the size $s$ of the BDD to $4 \cdot (k + (k - 1)) = 8k - 4$. Hence, each minimal-size BDD has to contain all cuts between two distinct blocks (these are $4 \cdot (k - 1)$) and exactly one cut in each block (these are $4 \cdot k$).

Furthermore, the two cuts in the $i$-th blocks of dimensions $d_P^\ell$ and $d_P^r$, or $d_Q^\ell$ and $d_Q^r$ correspond to a vertex selection in $P_i$, and $Q_i$, respectively. We achieve this as follows: Consider the $i$th block $T_{P,\ell}^i$ of $d_P^\ell$. Block $T_{P,\ell}^i$ (after the forcing examples and before the enforcing examples) contains an alternating sequence of badges (size more than $t$) of blue *separating examples* and badges (size more than $|E(G)|$ but less than $t$) of red *choice examples*. Each consecutive badge of separating examples and badge of choice examples can only be separated if the unique cut

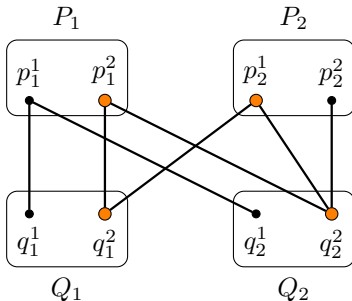

*Figure 4.* An instance of MULTICOLORED 2-BICLIQUEwith $n = 2$. A multicolored 2-biclique is depicted in orange.

between them in this block $T^i_{P,\ell}$ or in the corresponding block $T^i_{P,r}$ in dimension $d^r_P$ is part of the BDD. Our error threshold $t$ is chosen such that all separating examples have to be classified correctly and that exactly $2k$ badges of choice examples have to be classified correctly. Since each badge of separating examples has size more than $t$, at most one badge of choice examples of each block can be classified correctly. This badge of choice examples then corresponds to a vertex selection in $P_i$.

Additionally, we add one red *edge example* $e$ per edge in $G$ such that $e$ can only by correctly classified if both endpoints of the corresponding edge are selected. To achieve this, we add badges of blue *restricting examples* of size more than $t$ which have the effect that selecting one endpoint of an edge is not sufficient to classify this edge correctly as red. By setting $t$ accordingly, we ensure that the selected vertices from a multicolored $k$-biclique.

**Construction:** We first define the parameters $s$ and $t$, and then we describe the data set. A visualization is shown in Figures 4 to 7.

*Parameters:* Let $z = n^9$ and $x = n^5$. Both numbers are used to set the size of sets of examples. We set $s = 4k + 4(k-1) = 8k-4$, and we set $t = 2 \cdot k(n-1) \cdot x + |E(G)| - k^2$.

*Description of the examples:* By a *badge* we denote a set of indistinguishable examples, that is, all examples in that set have the same thresholds in all dimensions.

- We create a badge $B^i_{H,y}$ of blue *forcing examples* of size $z$ and a badge $R^i_{H,y}$ of red *enforcing examples* of size $z$ for each $H \in \{P,Q\}$, each $y \in \{\ell,r\}$, and each $i \in [k]$.

- We create an red *edge example* $e(u,w)$ for each edge $(u,w) \in E(G)$.

- We create a badge $C_u$ of red *choice examples* of size $x$ for each vertex $u \in V(G)$.

- We create a badge $S^a_{H,j}$ of blue *separating examples* of size $z$ for each $H \in \{P,Q\}$, $a \in [k]$ and each $j \in [n+1]$.

- We create a badge $F^j_u$ of blue *restricting examples* of size $z$ for each vertex $u \in V(G)$ and each $j \in [2k]$.

The following notation is useful for the assignment of thresholds to the examples: For each vertex $u \in V(G)$, we say that the badge $F^a_u$ of blue restricting examples for each $a \in [2k]$, the badge $C_u$ of red choice examples, and the red edge examples corresponding to all edges having one endpoint $u$, are the *examples associated with vertex* $u$. Observe that each edge example $e(u,w)$ is associated with both $u$ and $w$.

Note that $|V(G)| = 2 \cdot k \cdot n$. In total, we add $4 \cdot k \cdot z$ forcing examples, $4 \cdot k \cdot z$ enforcing examples, $m \leq |V(G)|^2 = 4 \cdot k^2 \cdot n^2$ edge examples, $|V(G)| \cdot x$ choice examples, $2 \cdot z \cdot k \cdot (n+1)$ separating examples, and $2k \cdot |V(G)| \cdot z$ restricting examples. Since $k \leq n$, these are $\mathcal{O}(n^{12})$ examples.

*Description of the dimensions:* We create 4 dimensions $d^\ell_P$, $d^r_P$, $d^\ell_Q$, and $d^r_Q$. In each dimension dim, the examples are arranged in $k$ *blocks* and one set Rest(dim), which contains all remaining examples. We next describe the $i$th block $T^i_{P,\ell}$ ($T^i_{P,r}$) of dimension $d^\ell_P$ ($d^r_P$), see also Figure 5 for an visualization. We describe the example sets at each threshold in increasing order.

- At the first threshold in the block $T^i_{P,\ell}$ we have the badge $B^i_{P,\ell}$ of blue forcing examples and the badge $F^i_q$ of blue restricting examples for any vertex $q \in Q$.

  At the first threshold in the block $T^i_{P,r}$ we have the badge $R^i_{P,r}$ of red enforcing examples and the badge $F^{k+1-i}_q$ of blue restricting examples for any vertex $q \in Q$.

- The next thresholds are populated alternatingly as follows: First, we have the badge $S^i_{P,1}$ of blue separating examples. Second, at the next threshold, we have all examples associated with the first vertex $p^1_i \in P_i$. Third, at the next threshold, we have the badge $S^i_{P,2}$ of blue separating examples. Fourth, we have all examples associated with the second vertex $p^2_i \in P_i$. This continuous until the badge $S^i_{P,n+1}$ of blue separating examples.

- At the last threshold in the block $T^i_{P,\ell}$ we have the badge $R^i_{P,\ell}$ of red enforcing examples and the badge $F^{k+i}_q$ of blue restricting examples for any vertex $q \in Q$.

  At the first last in the block $T^i_{P,r}$ we have the badge $B^i_{P,r}$ of blue forcing examples and the badge $F^{2k+1-i}_q$ of blue restricting examples for any vertex $q \in Q$.

$$T_{P,\ell}^i : \quad B_{P,\ell}^i \; F_Q^i \quad \bigg| \quad S_{P,1}^i \quad \bigg| \quad E(p_i^1) \; C_{p_i^1} \; F_{p_i^1} \quad \bigg| \quad S_{P,2}^i \quad \bigg| \quad E(p_i^2) \; C_{p_i^2} \; F_{p_i^1} \quad \bigg| \quad S_{P,3}^i \quad \bigg| \quad F_Q^{k+i} \; R_{P,\ell}^i$$

$$T_{P,r}^i : \quad R_{P,r}^i \; F_Q^i \quad \bigg| \quad S_{P,1}^i \quad \bigg| \quad E(p_i^1) \; C_{p_i^1} \; F_{p_i^1} \quad \bigg| \quad S_{P,2}^i \quad \bigg| \quad E(p_i^2) \; C_{p_i^2} \; F_{p_i^1} \quad \bigg| \quad S_{P,3}^i \quad \bigg| \quad F_Q^{k+i} \; B_{P,r}^i$$

*Figure 5.* Visualization of the two blocks $T_{P,\ell}^i$ and $T_{P,r}^i$ corresponding to the instance of Figure 4. Here, $E(p_i^j)$ refers to all red edge examples corresponding to edges having endpoint $p_i^j$. Furthermore, $F_Q^i = \{F_q^i : q \in Q\}$ and $F_Q^{k+i} = \{F_q^{k+i} : q \in Q\}$. A possible cut in a block is depicted as "|". The unique cut in each block of the optimal BDD shown in Figure 7 is shown in orange.

$$d_P^\ell : \quad T_{P,\ell}^1 \quad \bigg| \quad T_{P,\ell}^2 \quad \bigg| \quad \mathrm{Rest}(d_P^\ell)$$

$$d_P^r : \quad \mathrm{Rest}(d_P^r) \quad \bigg| \quad T_{P,r}^2 \quad \bigg| \quad T_{P,r}^1$$

$$d_Q^\ell : \quad T_{Q,\ell}^1 \quad \bigg| \quad T_{Q,\ell}^2 \quad \bigg| \quad \mathrm{Rest}(d_Q^\ell)$$

$$d_Q^r : \quad \mathrm{Rest}(d_Q^r) \quad \bigg| \quad T_{Q,r}^2 \quad \bigg| \quad T_{Q,r}^1$$

*Figure 6.* Visualization of the four dimensions using the blocks and the set of unused examples. The cuts "|" between each two consecutive blocks and between the last block and the set of unused examples has to be contained in every optimal BDD.

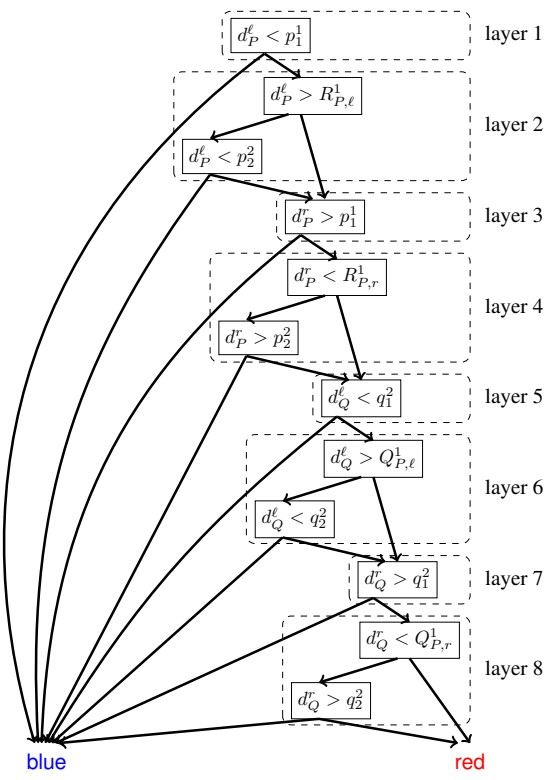

*Figure 7.* Visualization of the optimal BDD $B$ and its 8 layers corresponding to the instance of Figure 4 which is described in the ($\Rightarrow$) direction of the correctness proof.

Note that for each badge of forcing examples and each badge of enforcing examples in one of the $k$ blocks in dimensions $d_P^\ell$ and $d_P^r$, there exists exactly one badge of restricting examples $F_q^j$ for each $q \in Q$ at the same threshold.

The blocks $T_{Q,\ell}^i$ and $T_{Q,r}^i$ for dimensions $d_Q^\ell$ and $d_Q^r$ are defined analogously.

Now, we describe how the blocks $T_{P,\ell}^i$ and $T_{P,r}^i$ are arranged in dimensions $d_P^\ell$ and $d_P^r$, respectively. For this, we also need the *unused examples* $\mathrm{Rest}(d_P^\ell)$ in dimension $d_P^\ell$. Essentially, $\mathrm{Rest}(d_P^\ell)$ contains all examples which are not contained in any of the blocks $T_{P,\ell}^1, \ldots, T_{P,\ell}^k$. The sets of unused examples $\mathrm{Rest}(d_P^r)$, $\mathrm{Rest}(d_Q^\ell)$, and $\mathrm{Rest}(d_Q^r)$ are defined analogously. We refer to Figure 6 for a visualization.

In dimension $d_P^\ell$ we have the blocks $T_{P,\ell}^1, T_{P,\ell}^2, \ldots, T_{P,\ell}^k$ in that specific ordering, followed by the unused examples $\mathrm{Rest}(d_P^\ell)$. In dimension $d_P^r$ we first have the unused examples $\mathrm{Rest}(d_P^r)$, and then the blocks $T_{P,r}^k, T_{P,r}^{k-1}, \ldots, T_{P,r}^1$ in that specific ordering. The arrangement of the blocks and unused examples in the dimensions $d_Q^\ell$ and $d_Q^r$ is analogously.

**Correctness:** We show that $G$ has a multicolored $k$-biclique if and only if there exists a BDD $B$ with at most $s$ inner nodes making at most $t$ errors.

In the following, for a dimension $\dim \in \{d_P^\ell, d_P^r, d_Q^\ell, d_Q^r\}$ and a badge $X$ (or an example $e$), we use the notation $\dim < X$ ($\dim < e$), to denote a cut, that is, all examples which have a threshold smaller than the examples of badge $X$ (example $e$) in $\dim$ go to the left child of this cut, and all remaining examples go to the right child of this cut. Moreover, we say that a cut $\dim < X$ *violates* an example $e$ if $e[\dim] \geq X$, and otherwise we say that $\dim < X$ *applies to* $e$.

($\Rightarrow$) Let $S$ be a multicolored $k$-biclique of $G$, see Figure 4 for an example. By $p_i$ and $q_i$ we denote the unique vertex of $S$ in $P_i$ or in $Q_i$, respectively. We now design a BDD $B$ with exactly $s = 8k - 4$ inner nodes making at most $t$ errors, see also Figure 7 for an visualization. We want to emphasis that the constructed BDD $B$ is not unique, that is, there are also other BDD's with exactly $s$ inner nodes making at most

$t$ errors.

Basically, the BDD $B$ has $4k$ *layers*, see Figure 7 for a visualization. A layer is a sequence of (at most two) cuts leading to the blue leaf if all cuts of this layer apply to an example $e$, and otherwise, if at least one cut of this layer violates an example $e$, then $e$ is redirected into the next layer. Each example which is violated by at least one cut in the last layer is then put into the red leaf.

We say that layer $j$ *applies* to an example $e$ if $e$ can be put to the blue leaf via layer $j$. Otherwise, we say layer $j$ *violates* $e$. We use the following observation:

**Claim B.1.** (a) If all layers violate an example $e$, then $e$ is classified as red.

(b) Otherwise, if any of the $4k$ layers applies $e$, then $e$ is classified as blue.

Consequently, in order to show that a blue example $e$ is correctly classified it is sufficient that at least one of the layers applies to $e$. Furthermore, any red example $e$ can only be correctly classified by $B$ if all layers violate $e$.

The first $k$ layers include all cuts in dimension $d_P^\ell$. Moreover, layer $j$ corresponds to the block $T_{P,\ell}^j$. Furthermore, the second $k$ layers include all cuts in dimension $d_P^r$. Moreover, layer $(k+j)$ corresponds to the block $T_{P,r}^j$. Together these $2k$ layers correspond to the $k$ vertices of $S \cap P$. More precisely, layers $j$ and $(k+j)$ correspond to the vertex $p_j$ selected in the partite set $P_j$.

For the remaining $2k$ layers an analog property holds for dimensions $d_Q^\ell$ and $d_Q^r$. We now describe the first $k$ layers and then the second $k$ layers in detail:

*First $k$ layers:*

- Layer 1 only consists of the cut $d_P^\ell < p_1$.

- Layer 2 consists of the two cuts $d_P^\ell > R_{P,\ell}^1$ and $d_P^\ell < p_2$.

- Layer 3 consists of the two cuts $d_P^\ell > R_{P,\ell}^2$ and $d_P^\ell < p_3$.

- Layer $k$ consists of the two cuts $d_P^\ell > R_{P,\ell}^{k-1}$ and $d_P^\ell < p_k$.

*Layers $k+1$ to $2k$:*

- Layer $k+1$ only consists of the cut $d_P^r > p_1$.

- Layer $k+2$ consists of the two cuts $d_P^r < R_{P,r}^1$ and $d_P^r > p_2$.

- Layer $k+3$ consists of the two cuts $d_P^r < R_{P,r}^2$ and $d_P^r > p_3$.

- Layer $2k$ consists of the two cuts $d_P^r < R_{P,r}^{k-1}$ and $d_P^r > p_k$.

Layers $2k+1$ to $4k$ are defined analogously. We observe the following:

**Claim B.2.** If an example $e$ has a threshold in some block $T$ in some dimension $\dim$, then the layer corresponding to $T$ in $\dim$ is the unique layer in $\dim$ which can apply to $e$.

*Proof of Claim.* For simplicity assume without loss of generality that $\dim = d_P^\ell$ and that $T = T_{P,\ell}^i$. Any example $e \in T_{P,\ell}^i$ has a larger threshold than each vertex $p_j$ with $j < i$ in $d_P^\ell$. Consequently, the cut $d_P^\ell < p_j$ in layer $j$ violates $e$. Also, $e$ has a smaller threshold than each example in $R_{P,\ell}^j$ for each $j \in [i+2, k]$. Consequently, the cut $d_P^\ell$ in layer $j-1$ violates $e$. ∎

Claim B.2 implies that if $e \in \text{Rest}(\dim)$, then each layer in dimension $\dim$ violates $e$.

Clearly, the BDD $B$ consists of exactly $s$ inner nodes. Hence, it remains to verify that $B$ makes at most $t$ errors.

*$B$ makes at most $t = 2 \cdot k(n-1) \cdot x + |E(G)| - k^2$ errors.* Observe that $t < z = n^9$. We first verify that all forcing examples, all enforcing examples, all separating examples, and all restricting examples are correctly classified by $B$. Second, we show that exactly $2k$ badges of choice examples are correctly classified by $B$. Recall that each badge of choice examples has size $x$. Also, recall that we have $|V(G)| \cdot x$ choice examples in total. Thus, the BDD $B$ then misclassifies exactly $(|V(G)| - k) \cdot x - 2 \cdot k \cdot x = 2 \cdot k \cdot (n-1) \cdot x$ choice examples. Finally, we show that $B$ correctly classifies all edge examples corresponding to edges having both endpoints in $S$. Since $S$ is a multicolored $k$-biclique, these are $k^2$ examples. Recall that we have exactly $|E(G)|$ edge examples. Hence, if we have shown the above 3 steps, then we have verified that $B$ makes at most $t$ errors. We use Claims B.1 and B.2 to show these statements.

- *Forcing examples:* We show that for a badge $B_{P,\ell}^i$ of blue forcing examples the $i$th layer applies: The cut $d_P^\ell > R_{P,\ell}^{i-1}$ applies to $B_{P,\ell}^i$ since $B_{P,\ell}^i$ is contained in the next block $T_{P,\ell}^i$. Also the cut $d_P^\ell < p_i$ applies to $B_{P,\ell}^i$ since the block $T_{P,\ell}^i$ starts with the set $B_{P,\ell}^i$.

  Analogously, one can show that the $(k+i)$th layer applies to $B_{P,r}^i$, that the $(2k+i)$th layer applies to $B_{Q,\ell}^i$, and that the $(3k+i)$th layer applies to $B_{Q,r}^i$.

- *Enforcing examples:* Let $e \in R_{P,\ell}^i$. By Claim B.2, all layers except layer $i$ violate $e$. Also, layer $i$ violates $e$ since the cut $d_P^\ell < p_j$ violates $e$.

  Analogously, one can show that non of the layers applies for any other red enforcing example.

- *Separating examples:* Consider the badge $S_{P,j}^i$ of blue separating examples. Recall that $\{p_i\} = S \cap P_i$. By $h \in [n]$ we denote the index of $p_i$ in the ordering $p_i^1, p_i^2, \ldots, p_i^n$, that is $p_i = p_i^h$. We show that either (a) the $i$th layer applies if $j \leq h$, or (b) the $(k+i)$th layer applies if $j > h$. Without loss of generality we assume that $j \leq h$. The $i$th layer applies since the cut $d_P^\ell > R_{P,\ell}^{i-1}$ applies to $S_{P,j}^i$ since $S_{P,j}^i$ is contained in the next block $T_{P,\ell}^i$. Also the cut $d_P^\ell < p_i$ applies to $S_{P,j}^i$ since $j \leq h$ and $p_i = p_i^h$.

  Analogously, one can find a layer which applies to each other separating example.

- *Restricting examples:* We show that for the badge $F_u^i$ of blue restricting examples where $u \in Q$ and $i \in [k]$ the $i$th layer applies: The cut $d_P^\ell > R_{P,\ell}^{i-1}$ applies to $F_u^i$ since $F_u^i$ is contained in the next block $T_{P,\ell}^i$. Also the cut $d_P^\ell < p_i$ applies to $F_u^i$ since the block $T_{P,\ell}^i$ starts with the set $F_u^i$.

  Analogously, one can show that the $(2k+i)$th layer applies to the blue restricting examples $F_u^i$ where $u \in Q$ and $u \in [k+1, 2k]$. Then, the argumentation for $F_u^i$ with $u \in P$ is analogously.

- *Choice examples:* Consider the badge $C_{p_i}$ of red choice examples. Recall that $\{p_i\} = S \cap P_i$. By Claim B.2 we only need to consider layers $i$ and $(k+i)$. Layer $i$ violates $C_{p_i}$ since the cut $d_P^\ell < p_i$ violates $C_{p_i}$, and layer $k+i$ violates $C_{p_i}$ since the cut $d_P^r > p_i$ violates $C_{p_i}$.

  Analogously, one can show that all layers violate the red choice examples $C_{q_i}$ where $\{q_i\} = S \cap Q_i$.

- *Edge examples:* Let $e = e(p_i, q_j)$ be an red edge example corresponding to an edge $(p_i, q_j) \in E(G)$ such that $p_i, q_j \in S$. The argumentation is almost identical to the choice examples $C_{p_i}$ with $p_i \in S$: By Claim B.2 we only need to consider layers $i$, $(k+i)$, $(2k+j)$, and $(3k+j)$. Layer $i$ violates $e$ since the cut $d_P^\ell < p_i$ violates $e$, layer $(k+i)$ violates $e$ since the cut $d_P^r > p_i$ violates $e$, layer $(2k+j)$ violates $e$ since the cut $d_Q^\ell < q_j$ violates $e$, and layer $(3k+j)$ violates $e$ since the cut $d_Q^r > q_j$ violates $e$.

($\Leftarrow$) Let $B$ be a BDD with at most $s$ inner nodes making at most $t$ errors. We now show that $G$ has a multicolored $k$-biclique.

*Outline:* First, we show that $B$ has exactly $s = 8k - 4$ inner nodes. Note that we only argue about which cuts the BDD $B$ has to contain; the structure of $B$ is not important. More precisely, we show that $B$ contains all unique cuts which splits all blocks in all dimensions (these are exactly $4 \cdot (k-1) = 4k - 4$ many) and exactly one cut within

each block (these are exactly $4 \cdot k$ many). We achieve this since all blue forcing examples have to be separated from all red enforcing examples by $B$ since both the forcing and the enforcing examples are arranged in badges of $z > t$ examples each. Second, we verify that the cuts in blocks $T_{P,\ell}^i$ and $T_{P,r}^i$ have the form $d_P^\ell < p_i^j$ or $d_P^\ell \leq p_i^j$ and $d_P^r > p_i^j$ or $d_P^r \geq p_i^j$ for some vertex $p_i^j \in P_i$. An analog property also holds for the blocks $T_{Q,\ell}^i$ and $T_{Q,r}^i$. Let $S$ be the corresponding vertex set. Finally, we verify that $S$ corresponds to a multicolored $k$-biclique. Clearly, $S$ contains exactly one vertex of each partite set, and thus we only have to show that there exists an edge between any vertex in $S \cap P$ and any vertex in $S \cap Q$.

*Step 1:* First, we show that $B$ contains all cuts between two consecutive blocks in all 4 dimensions. Recall that these are $4 \cdot (k-1) = 4k - 4$ many. Without loss of generality, we consider the two consecutive blocks $T_{P,\ell}^i$ and $T_{P,\ell}^{i+1}$. Now, consider the badge $R_{P,\ell}^i$ of red enforcing examples of block $T_{P,\ell}^i$ and the badge $B_{P,\ell}^{i+1}$ of blue forcing examples of block $T_{P,\ell}^{i+1}$. Note that $d_P^\ell < B_{P,\ell}^{i+1}$ is the unique cut in dimension $d_P^\ell$ which separates $R_{P,\ell}^i$ from $B_{P,\ell}^{i+1}$. Furthermore, we have $R_{P,\ell}^i, B_{P,\ell}^{i+1} \in \text{Rest}(\dim)$ for each remaining dimension $\dim \in \{d_P^r, d_Q^\ell, d_Q^r\}$. Consequently, $B$ contains the unique cut $d_P^\ell < B_{P,\ell}^{i+1}$ between the two blocks $T_{P,\ell}^i$ and $T_{P,\ell}^{i+1}$.

Second, we verify that $B$ contains at least one cut within each block. Since we have $4k$ blocks and $s \leq 8k - 4$, we can then conclude that $B$ contains exactly one cut within each block. Without loss of generality, we consider the block $T_{P,\ell}^i$. Now, consider the badge $B_{P,\ell}^i$ of blue forcing examples and the badge $R_{P,\ell}^i$ of red enforcing examples of this block $T_{P,\ell}^i$. Furthermore, we have $B_{P,\ell}^i, R_{P,\ell}^i \in \text{Rest}(\dim)$ for each remaining dimension $\dim \in \{d_P^r, d_Q^\ell, d_Q^r\}$. Consequently, $B$ contains at least one cut within the block $T_{P,\ell}^i$.

*Step 2:* By construction, all blue examples and also all red enforcing examples are arranged in badges of size $z > t$ each. Consequently, all blue examples and also all red enforcing examples have to be classified correctly by $B$. Moreover, observe that all red choice examples are arranged in badges of size $x = n^5$ each, and that we have $|E(G)| < (kn)^2 < n^5$ red edge examples. Since $t = 2 \cdot k(n-1) \cdot x + |E(G)| - k^2$, we conclude that $B$ has to classify at least $2k$ badges of choice examples correctly.

We now show that $B$ can classify *at most* one badge of choice example per partite set $P_i$ or $Q_i$ correctly. Since there are exactly $2k$ partite sets, we can then conclude that $B$ classifies *exactly* one badge of choice example per partite set correctly.

It remains to verify that $B$ can classify at most one badge of choice example per partite set, say $P_1$, correctly. Consider the badge $C_{p_1^j}$ of red choice examples corresponding to the $j$th vertex of $P_1$ and also the two badges $S_{P,j}^1$ and $S_{P,j+1}^1$ of blue separating examples. By construction, $C_{p_1^j}, S_{P,j}^1, S_{P,j+1}^1$ are contained in the blocks $T_{P,\ell}^1$ and $T_{P,r}^1$, and in $\text{Rest}(d_Q^\ell)$ and $\text{Rest}(d_Q^r)$. Moreover, we observe the following:

**Claim B.3.** (a) $C_{p_1^j}$ and $S_{P,j}^1$ can only be separated by the cuts $d_P^\ell < p_1^j$ and $d_P^r \geq p_1^j$, and (b) $C_{p_1^j}$ and $S_{P,j+1}^1$ can only be separated by the cuts $d_P^\ell \leq p_1^j$ and $d_P^r > p_1^j$.

By Step 1, we know that $B$ contains exactly one cut in each block. Thus, both cuts in blocks $T_{P,\ell}^1$ and $T_{P,r}^1$ are required to classify $C_{p_1^j}$ correctly. Moreover, for any other vertex $p_1^h \in P_1$, the two necessary cuts to classify $C_{p_1^h}$ correctly are disjoint from the necessary cuts for $C_{p_1^j}$. Consequently, $B$ can classify at most one badge of choice example per partite set correctly.

Hence, we have now verified that $B$ classifies exactly one badge of choice examples per partite set correctly. Since each badge of choice examples corresponds to a vertex of $V(G)$, we obtain a vertex set $S$ containing exactly one vertex per partite set $P_i$ and $Q_i$.

*Step 3:* According to our choice of $t = 2 \cdot k(n-1) \cdot x + |E(G)| - k^2$ and to Step 2, we conclude that $B$ has to correctly classify at least $k^2$ red edge examples. In the following, we show that any edge example $e$ corresponding to an edge $(u,w)$ such that at least one of its endpoints is not contained in $S$, is misclassified by $B$. If we have verified this, we can then conclude that $S$ is a multicolored $k$-biclique.

Now, let $e = e(u,w)$ be an red edge example such that at least one of the endpoints of the corresponding edge $(u,w)$, say $u$, is not contained in $S$. Without loss of generality, we assume that $u = p_1^i \notin S$. Now, let $\{p_1^j\} = S \cap P_1$. Note that $i \neq j$. First, we consider the case that $i < j$, see also Figure 5 for a visualization. Consider the badge $F_w^1$ of blue restricting examples. Observe that both $e$ and $F_w^1$ are contained in the blocks $T_{P,\ell}^1$ and $T_{P,r}^1$. Recall that by Claim B.3, the unique cut in dimension $d_P^\ell$ is either $d_P^\ell < p_1^j$ or $d_P^\ell \leq p_1^j$ and that the unique cut in dimension $d_P^r$ is either $d_P^r \geq p_1^j$ or $d_P^r > p_1^j$. Consequently, the unique cut of $B$ in $T_{P,\ell}^1$ applies to both $e$ and $F_w^1$, and the unique cut of $B$ in $T_{P,r}^1$ violates by both $e$ and $F_w^1$. Moreover, $e$ and $F_w^1$ use the identical thresholds in $d_Q^\ell$ and $d_Q^r$. Thus, we conclude that $e$ and $F_w^1$ cannot be distinguished by $B$. By construction, $|F_w^1| = z > t$ and thus both $e$ and $F_w^1$ are classified as blue. Second, the case that $i > j$ can be

handled analogously by considering the set $F_w^{k+1}$. Hence, we have shown that $e$ gets misclassified by $B$.

**Lower Bound:** Recall that in the constructed instances, $d = 4$. Since MULTICOLORED $k$-BICLIQUE is NP-hard and the reduction runs in polynomial time, we obtain the claimed NP-hardness. Moreover, since $s = 8k - 4$ in the constructed instances and MULTICOLORED $k$-BICLIQUE cannot be solved in $f(k) \cdot |V(G)|^{o(k)}$ time unless the ETH fails (Cygan et al., 2015), we obtain that ERROR BSBDD cannot be solved in $f(s + d) \cdot |I|^{o(s+d)}$ time if the ETH is true, where $|I|$ is the overall instance size, even if $d = 4$.

Observe that the BDD constructed in the proof of Theorem 4.1 is also an OBDD. Hence, we obtain the same hardness results also for ERROR BSOBDD. $\square$

### B.2. Proof of Theorem 4.2

*Proof.* Initially, we bound the size of any minimal decision tree for the corresponding classification instance $(E, \lambda)$ of the ERROR BSBDD or ERROR BSOBDD instance. Since decision tree is also a BDD, this number is then an upper bound for any minimal size BDD. This upper bound then allows us to brute force the size of an optimal BDD. Afterwards, we can use the same ideas of the XP-algorithm for $s$ behind Theorem 3.1 to first guess the structure of the BDD and second to populate the inner nodes with cuts.

Note that in each minimal-size decision tree no leaf is empty, that is, each leaf contains at least one example. Consequently, the length of each root-leaf path is bounded by the maximal number of cuts which is $d \cdot D$. In other words, the depth of each minimum decision tree is at most $d \cdot D$. Since decision trees are binary trees, each minimal size decision tree contains at most $2^{d \cdot D}$ inner nodes. Consequently, each minimal size BDD has size at most $2^{d \cdot D}$.

Now, we can guess the size $z \in [2^{d \cdot D}]$ of a minimum-size BDD. Next, for this size $z$, we use the algorithm behind Theorem 3.1. In other words, first, in $z^{2z} \cdot |I|^{\mathcal{O}(1)}$ time, we guess the structure of the BDD, where $|I|$ is the overall instance size. Second, in $z^{d \cdot D} \cdot |I|^{\mathcal{O}(1)}$ time, we guess the cut for each node in the BDD. For OBBDs, we additionally check whether the order property is fulfilled. Overall, the running time is $(2^{d \cdot D})(2^{d \cdot D+1} + d \cdot D) \cdot \text{poly}(|I|)$, where $|I|$ is the overall instance size. The algorithm is correct since a minimal BDD $B$ has size $z \leq 2^{d \cdot D}$ and at one stage we try this specific value of $z$. Moreover, since we try all possibilities for the structure of the BDD, the algorithm also finds the structure of $B$ and since we try all possibilities for all cuts, the algorithm also finds the correct cuts. $\square$

### B.3. Proof of Proposition 4.3

*Proof.* (1) We reduce from HITTING SET. The input consists of an universe $\mathcal{U}$, a family of subsets $\mathcal{F}$ of $\mathcal{U}$ and a number $k$. The goal is to select $S \subseteq \mathcal{U}$ of size $k$ such that each set $F \in \mathcal{F}$ contains at least one element of $S$. HITTING SET is NP-hard and cannot be solved in $f(k) \cdot n^{o(k)}$ time, unless the ETH fails (Cygan et al., 2015).

We create a classification instance as follows: For each element $u \in \mathcal{U}$ we create a binary dimension $d_u$. Furthermore, for each set $F \in \mathcal{F}$ we create a blue example $e_F$. Example $e_F$ has value $1$ in each dimension corresponding to an element $u$ contained in $F$, and value $0$ in each other dimension. Afterwards, we add a red example $e^*$ which has value $0$ in each dimension. Clearly, $D = 2$. Finally, we set $s = k$

Note that each minimal size decision tree is equivalent to a minimal size hitting set (Ordyniak & Szeider, 2021). Since the classification instance contains exactly one red example $e^*$, each minimal size decision tree is a path such that all leaves are blue except one leaf of the last cut which is red. Consequently, each minimal size BDD has the same structure. Furthermore, since the cuts in each such path can be reordered arbitrarily without misclassifying any example, this result extends to OBBDs.

(2) This result follows by the same reduction by reducing from the NP-hard VERTEX COVER which is the special case of HITTING SET where each set in the family $\mathcal{F}$ has size $2$. As a consequence, we obtain $\delta = 2$. $\qquad\square$

## C. Additional Material for Section 5

We improved the SAT-encoding of Cabodi et al. (2024) for the following reason: It is not clear to us how the $\rightarrow$ implication of the $\leftrightarrow$ constraint 19) of their model can be encoded efficiently without substantially increasing the number of used variables. This constraint models the classification path of an example $m$. However, for the correctness of the model it is sufficient to have the $\leftarrow$ implication in constraint 19): Initially, $m$ is contained in the root $i$ and since the root has a dimension $k$, and a left and right child, example $m$ is then redirected to exactly one of them, say $n$. This ensures that the right hand side for these values of $i, n, m, k$ is fulfilled and thus $e_{mn} = 1$. Applying this argument iteratively leads to the classification path of example $m$.

| Instance name | $n$ | $n'$ | $d$ | $d'$ | $c$ | $c'$ | $\delta$ | $\delta'$ | $D$ | $D'$ |
|---|---|---|---|---|---|---|---|---|---|---|
| postoperative-patient-data | 72 | 72 | 22 | **17** | 22 | 22 | 14 | 14 | **2** | 3 |
| hayes-roth | 84 | 84 | 15 | 15 | 15 | 15 | 8 | 8 | 2 | 2 |
| lupus | 86 | **79** | 3 | **2** | 126 | **78** | 3 | **2** | 75 | **53** |
| appendicitis | 106 | 106 | 7 | 7 | 523 | **460** | 7 | 7 | 99 | **98** |
| molecular_biology_promoters | 106 | 106 | 228 | 228 | 228 | 228 | 104 | 104 | 2 | 2 |
| tae | 106 | 106 | 5 | 5 | 96 | **94** | 5 | 5 | 46 | **45** |
| cloud | 108 | 108 | 7 | 7 | 585 | **555** | 7 | 7 | 108 | **100** |
| cleveland-nominal | 130 | 130 | 17 | 17 | 17 | 17 | 11 | 11 | 2 | 2 |
| lymphography | 148 | 148 | 50 | **37** | 50 | 50 | 26 | **23** | **2** | 3 |
| hepatitis | 155 | 155 | 39 | **28** | 355 | **335** | 28 | **25** | 85 | 85 |
| glass2 | 162 | 162 | 9 | 9 | 709 | **667** | 9 | 9 | 136 | **132** |
| backache | 180 | 180 | 55 | **50** | 469 | **429** | 26 | 26 | 180 | **151** |
| auto | 202 | 202 | 52 | **35** | 961 | **916** | 31 | **29** | 184 | **182** |
| glass | 204 | 204 | 9 | 9 | 894 | **846** | 9 | 9 | 172 | **165** |
| biomed | 209 | 209 | 14 | 14 | 735 | **577** | 9 | 9 | 191 | **125** |
| new-thyroid | 215 | **214** | 5 | 5 | 329 | **232** | 5 | 5 | 100 | **73** |
| spect | 219 | 219 | 22 | 22 | 22 | 22 | 22 | 22 | 2 | 2 |
| breast-cancer | 266 | 266 | 31 | **25** | 40 | 40 | 15 | 15 | 11 | 11 |
| heart-statlog | 270 | 270 | 25 | **23** | 376 | **369** | 18 | 18 | 144 | **142** |
| haberman | 283 | 283 | 3 | 3 | 89 | **86** | 3 | 3 | 49 | **46** |
| heart-h | 293 | 293 | 29 | **22** | 325 | **318** | 19 | 19 | 154 | 154 |
| hungarian | 293 | 293 | 29 | **22** | 325 | **318** | 19 | 19 | 154 | 154 |
| cleve | 302 | 302 | 27 | **25** | 390 | **382** | 18 | 18 | 152 | **151** |
| heart-c | 302 | 302 | 27 | **25** | 390 | **382** | 18 | 18 | 152 | **151** |
| cleveland | 303 | 303 | 27 | **25** | 391 | **383** | 18 | 18 | 152 | **151** |
| ecoli | 327 | **326** | 7 | **5** | 351 | **233** | 6 | **5** | 81 | **59** |
| schizo | 340 | 340 | 14 | 14 | 2218 | **2209** | 14 | 14 | 203 | 203 |
| bupa | 341 | 341 | 5 | 5 | 307 | **302** | 5 | 5 | 94 | 94 |
| colic | 357 | 357 | 75 | **71** | 408 | **400** | 36 | 36 | 85 | **82** |
| dermatology | 366 | 366 | 129 | **101** | 188 | 188 | 57 | **53** | 61 | 61 |
| cars | 392 | **388** | 12 | **11** | 704 | **531** | 9 | 9 | 346 | **266** |
| soybean | 622 | 622 | 133 | **73** | 133 | **108** | 68 | **49** | **2** | 7 |
| australian | 690 | 690 | 18 | **16** | 1155 | **1119** | 16 | **15** | 350 | 350 |
| diabetes | 768 | 768 | 8 | 8 | 1246 | **1238** | 8 | 8 | 517 | **515** |
| contraceptive | 1358 | 1358 | 21 | 21 | 66 | **65** | 13 | 13 | 34 | 34 |

*Table 1.* Overview of the data sets we used for our experiments including their name, number of examples $n$, number of dimensions $d$, number of total cuts $c$, maximum number $\delta$ of dimensions in which two examples differ, and the largest domain size $D$. The columns $n', d', c', \delta'$, and $D'$ show the values of the data sets after applying all reduction rules described by Staus et al. (2025, Appendix B). **Bold** entries indicate a change of this specific value in the input instance and after the application of all reduction rules.

