# OpenReview forum: "Learning Minimum-Size BDDs: Towards Efficient Exact Algorithms"
_ICML.cc/2025/Conference — ICML 2025 poster_

### Official Review · Reviewer_Uo3X · 2025-02-18

**Overall Recommendation:** 3

**Summary:**

This is a paper about BDDs, intended as a more compact formulation of decision trees in the context of classification tasks. Komusiewicz et al. (2023) introduced the "witness" concept for decision trees, which is extended here to BDDs. The authors focus on the problem of deciding whether a BDD of given maximum size can exactly cover/classify a dataset. An algorithm for this (NP-hard) task is derived and shown to improve the sota. The algorithm can be employed as a subroutine for variants of the decision problem (e.g., partial coverage). The authors also present experiments where their algorithm is tested against a SAT-based approach to the problem. The proposed solution is much faster when small diagrams can cover the data, while the situation changes for medium and large models.

**Claims And Evidence:**

The paper's central claim is Theorem 3.3, and its proof is well detailed in the (body of the) paper.

In the final outlook section, the authors claim that their algorithm has the potential for further improvement, ideally leading to more competitive results with larger diagrams. Future work will determine whether this is the case.

**Essential References Not Discussed:**

I think all the relevant and recent literature in the field has been properly cited and considered for the discussion.

**Ethical Review Concerns:**

No concerns.

**Experimental Designs Or Analyses:**

The experiment's design and results analysis are relatively straightforward, but the approach seems fair for validating the new algorithm.

**Methods And Evaluation Criteria:**

The authors also provide a numerical validation against a SAT approach. This is very useful for adequately evaluating the algorithm's potential, and the selected benchmark looks sufficient to identify the proposed approach's pros (and cons).

**Other Comments Or Suggestions:**

s >=4 should be <=

**Other Strengths And Weaknesses:**

The contribution is evident, and the advantages to the SOTA are measurable. Applying the witness idea to this formalism is not trivial, and the result looks helpful.

The weakness is also evident. The algorithm is competitive only for small models.

**Questions For Authors:**

As the limitation of the paper is related to the size of the model that can be reasonably processed, the main question to the authors should be whether they believe that the s<=4 limitation should be intended as an intermediate step towards a further improvement of the procedure or not. In case of a positive answer, it would be essential to understand why the authors are convinced that the procedure could become faster and be used for larger models. In the case of a negative answer, convincing the reader that even the current limitation could make the procedure applicable for actual machine learning applications would be essential.

**Relation To Broader Scientific Literature:**

This paper is focused on a particular field of machine learning. I didn't miss references to broader literature.

**Theoretical Claims:**

The proofs of the results look sound and quite readable. My check was only superficial, but I did not spot any flaw.

---

> ### Author Rebuttal · Authors · 2025-03-31
>
> Thanks for your review!
>
> > As the limitation of the paper is related to the size of the model that can be reasonably processed, the main question to the authors should be whether they believe that the s<=4 limitation should be intended as an intermediate step towards a further improvement of the procedure or not.
>
> First, note that our solver WitBDD is not limited to instances where $s \le 4$, but also solves some instances where $s=7$. Furthermore, even in its current proof-of-concept state, already WitBDD would contribute to a portfolio solver for the case of small $s$. But to address the question, indeed, we believe there is large a potential for improvement and algorithm-engineering work, which we deem to be out of scope for the current paper: Our main focus is to develop the algorithmics of computing BDDs. Similar to decision trees, a first paper (Komusiewicz et al., ICML 2023) provided the theoretical foundations for an efficient algorithm and in a later algorithm-engineering project this algorithm was lifted to a competitive solver (Staus et al., 2024). Concretely, since WitBDD is based on branch-and-bound, symmetry-breaking techniques and improved lower bounds should give substantial speed-ups. However, this type of work would require a different presentation and set-up of the paper and would require much more content to be fitted hence we think this is out of scope for the present work.
>
> > In case of a positive answer, it would be essential to understand why the authors are convinced that the procedure could become faster and be used for larger models. In the case of a negative answer, convincing the reader that even the current limitation could make the procedure applicable for actual machine learning applications would be essential.
>
> See, above, we believe there is large potential for speed-ups. Furthermore, in general, it is useful to validate heuristics on ground truths, for which we need efficient exact algorithms such as those we provide. Second, exploring the exact algorithmics of a problem often reveals structure that can also be exploited heuristically as well.
>
> > "s >=4 should be <="
>
> Thanks!

---

### Official Review · Reviewer_kRbE · 2025-03-08

**Overall Recommendation:** 5

**Summary:**

The paper presents a novel approach for generating classification Binary Decision Diagrams (BDDs) of bounded size using the witness paradigm. Specifically, given a labeled dataset, the BDD is constructed through a branch-and-bound mechanism that incrementally refines an initial graph. This refinement process involves identifying a data point that is incorrectly classified by the initial BDD, modifying the graph by adding nodes and edges, and associating the newly added edge with the data point (the "witness") to reduce the search space. The paper also develops complexity results and provides experimental comparisons against a SAT encoding.

## update after rebuttal
Thank you for the rebuttal - it addressed my questions and I will keep my score as is.

**Claims And Evidence:**

The paper makes a strong contribution to the broad field of BDDs, and is mainly supported by a nice theoretical study. In particular, the proposed construction mechanism is efficient if the data points are "sufficiently" close, which is intuitive in the sense that it avoids extra branching. The numerical results also show promising results against one of the state-of-the-art compilation techniques, which I found to be another good evidence to the claims. Finally, I would like to highlight that the error analysis (Section 4) is shot but particularly interesting, and could serve as a foundation for future research streams in misclassification errors.

**Essential References Not Discussed:**

N/A

**Experimental Designs Or Analyses:**

The experiments are short and are more illustrative of the possible computational benefits of the theoretical approach studied here. I do believe they are sufficient in view of the theoretical scope of the paper.

**Methods And Evaluation Criteria:**

Although the proofs can be read as intricate, the methodology is actually intuitive: if a miss-classified point / counter example is found, the procedure applies a so-called one-step refinement that recursively inserts leaves (nodes with out-degree zero) to recover a notion of "consistency" relating witnesses with their expected classification paths.

As the authors mentioned, this technique is fundamentally a BDD extension of the work by Komusiewicz et al. (2023), who originally applied similar refinement principles to decision trees. The complexity results are also relatively comparable. However, this paper is by no means an incremental work, as there are fundamental structural differences between decision trees and BDDs (e.g., the one-step refinement here differs significantly in how leaves are generated). Instead, I see this as an intriguing example of how insights from the decision-tree literature can be extended to improve complexity results and advance the state-of-the-art in BDDs.

**Other Comments Or Suggestions:**

N/A

**Other Strengths And Weaknesses:**

N/A

**Questions For Authors:**

- Suppose there are multiple "dirty" witnesses that can be chosen during Algorithm 1. Does the sequence of witnesses to be considered influence the procedure in terms of feasibility/final size?

**Relation To Broader Scientific Literature:**

(See above.)

**Theoretical Claims:**

I read all proofs. I have not identified any particular issues, but they are generally significantly intricate because they rely on multiple constructive arguments. My general suggestion is to add further examples/figures in the proof to represent many of these constructive parts; e.g., in Claims C.1. and C.2 in the Supplemental Material, it would be great to connect the results more with Figure 7, which was very helpful to understand the underlying cuts and oeprations.

---

> ### Author Rebuttal · Authors · 2025-03-31
>
> Thank you for your review!
>
> > Suppose there are multiple "dirty" witnesses that can be chosen during Algorithm 1. Does the sequence of witnesses to be considered influence the procedure in terms of feasibility/final size?
>
> The algorithm will always find a minimum-size perfect BDD independent of which specific dirty example is chosen in any call to Algorithm 1. However, the dirty example affects the size of the algorithm's search space. We exploit this property by choosing dirty examples which we deem to yield a small search space, see Appendix D.1 for details.

---

### Official Review · Reviewer_2cDX · 2025-03-10

**Overall Recommendation:** 2

**Summary:**

This paper proposes a method for learning binary decision diagrams (BDD) that classify given training examples. The proposed algorithm finds the minimum-size BDD that can perfectly classify given examples. Starting from an empty BDD, the proposed algorithm repeats one-step refinements to update the structure of BDD and finds the minimum-size BDD that can correctly classify given samples. The paper theoretically analyses the correctness and the running time of the paper.  In experiments, the paper shows that the proposed method runs faster than the baseline method using a SAT solver when the size of the output BDDs is small.

### Update after rebuttal
I thank the authors for their reply. I agree that this is an interesting theoretical contribution. However, I still feel that the paper's impact is limited since the proposed algorithm currently scales for small-size BDDs. In such cases, the merits of using BDDs instead of decision trees will be limited.

I agree with the authors that algorithm engineering would further improve the scalability of the exact algorithm. However, we need more evidence to evaluate the practical potential of this paper.

**Claims And Evidence:**

The paper gives theoretical analyses to show that the proposed algorithm can find a minimum-size BDD. Moreover, it runs experiments to show that the proposed algorithm runs faster than the baseline method using a SAT solver.

**Essential References Not Discussed:**

N/A

**Experimental Designs Or Analyses:**

The experimental results are weak in supporting the proposal's effectiveness. The results show that WitBDD is faster than the baseline when BDD sizes are very small. The decision rules that can be represented with such a small BDD are simple ones. Therefore, the paper's results have limited impact in practical situations.

Historically, BDDs are used when binary decision trees are very large. If the minimum BDD sizes are smaller than 4, then the size of a decision tree representing the equivalent decision rules is also not large. Therefore, we do not have a strong motivation to use BDD in such cases.

**Methods And Evaluation Criteria:**

The proposed approach to finding a minimum BDD involves an exhaustive search. Since finding a minimum BDD is known to be NP-complete, I think this approach makes sense if we want an exact method.

**Other Comments Or Suggestions:**

- (Line 110) It seems strange to define a set of examples $E$ as a subset of $\mathbb{R}^d$.  How can we define the cardinality $|E|$?

**Other Strengths And Weaknesses:**

The paper seems clearly written and easy to read.

**Questions For Authors:**

N/A

**Relation To Broader Scientific Literature:**

The paper shows an algorithm for finding the minimum BDD classifying given examples perfectly. This might be a new result. However, its impact is limited since the proposed algorithm works only when the minimum BDD is small.

Compared with decision trees, BDDs are far less used in machine learning. Therefore, the impact of the results is limited.

**Theoretical Claims:**

I briefly checked the proof of the theoretical claims. They seem correct.

---

> ### Author Rebuttal · Authors · 2025-03-31
>
> Thanks for your review!
>
> The main purpose of our work is to further develop the algorithmics of computing BDDs, including providing algorithms and proving their correctness. The goal of our experiments is to provide a proof-of-concept implementation of our new approach and to compare it against the SOTA solvers. Doing algorithm engineering to achieve a very fast implementation is beyond the scope of this paper (already most proofs and implementation details are in the appendix).
> Similar to decision trees, a first paper (Komusiewicz et al., ICML 2023) provided the theoretical foundations for an efficient algorithm and in a later algorithm-engineering project this algorithm was lifted to a competitive solver (Staus et al., 2024).
> In this second paper, many heuristic lower bounds and reduction rules were provided. Showing the correctness of these lower bounds and rules required some effort.
> Similar to Staus et al., 2024 lots of promising improvements (symmetry breaking, improved lower bounds) are possible for WitBDD which then could yield a new SOTA solver.
> Moreover, even our proof-of-concept implementation can already contribute to a portfolio solver for the case of small $s$.
>
> Experimental Designs Or Analyses:
> 1) There is an overlapping use case for decision trees and BDDs. BDDs are not as commonly used as decision trees, however, one reason for this may be that not too many efficient algorithms for computing them are known. With this work, we want to address this gap. Consequently, our work should be understood as a first step towards having efficient implementations for computing optimal BDDs which then can have a huge impact.
>
> 2) Note that our solver WitBDD is not limited to instances where $s \le 4$, but also solves some instances where $s=7$.
>
> Other Comments and Suggestions:
> In our setting, $E$ is always a finite set. Hence, the cardinality is just the number of elements.

---

> > ### Comment · Reviewer_2cDX · 2025-04-04
> >
> > Thank you for the response. The paper's theoretical contributions are interesting. However, I still think that an exact algorithm that works well with BDDs of small sizes ($s \leq 7$) is practically weak because the sizes of decision trees and BDDs are not so different at this scale.
> >
> > > There is an overlapping use case for decision trees and BDDs. BDDs are not as commonly used as decision trees, however, one reason for this may be that not too many efficient algorithms for computing them are known. With this work, we want to address this gap. Consequently, our work should be understood as a first step towards having efficient implementations for computing optimal BDDs which then can have a huge impact.
> >
> > This is an interesting research direction. Is there any practical evidence that the proposed algorithm can contribute to realizing a scalable BDD learning algorithm?

---

> > > ### Author Response · Authors · 2025-04-08
> > >
> > > Thanks for your reply!
> > >
> > > > However, I still think that an exact algorithm that works well with BDDs of small sizes () is practically weak because the sizes of decision trees and BDDs are not so different at this scale.
> > >
> > > While there might not be a large difference between the size of BDDs and decision trees at the current scale, for (only slightly) larger s it is unclear whether this is still the case. To figure this out it is necessary to have better exact algorithms.
> > >
> > > > Is there any practical evidence that the proposed algorithm can contribute to realizing a scalable BDD learning algorithm?
> > >
> > > Indeed, we think there is some strong practical evidence:
> > >
> > > 1. Our implementation shows that our approach is feasible even without dedicated algorithm engineering, which is not always given when transferring theory into practice.
> > >
> > > 2. A single algorithm-engineering paper for decision trees (Staus et al. 2025) improved the naive implementation based on a similar witness-based algorithmic concept with a mean 324-fold speedup, yielding a state-of-the-art exact solver.
> > >
> > > 3. Generally this line of research of starting with a proof-of-concept exact algorithm and adding algorithm-engineering is highly effective.
> > > For instance, consider the CDCL algorithm for SAT which almost didn't change since its introduction in the late 90s; the current success (see https://cca.informatik.uni-freiburg.de/satmuseum/satmuseum2022.pdf) of CDCL is mainly explained by subsequent and recent algorithm engineering (note that the original/first CDCL implementation (grasp) performs even worse than DPLL (boehm) and only with subsequent algorithm engineering it was substantially faster (chaff)).
> > > Another example is the Concorde Traveling-Salesperson solver where the baselines is only feasible for small instances but a version with engineering can solve large practical problem instances.

---

### Official Review · Reviewer_cY6R · 2025-03-12

**Overall Recommendation:** 4

**Summary:**

The paper studies the decision problem of Bounded-Size Binary Decision Diagram (BSBDD). Given a labelled data set and a positive integer $s$, the algorithm asks if there exists a BDD that perfectly classifies each example in the labelled data set (each leaf node of the BDD represent a class), and has at most $s$ internal nodes.

The main result is an algorithm `WitBDD`  that takes as input $s$, a WBDD (witness BDD, a special representation of a BDD) with less than $s$ internal nodes $W$, and a labelled dataset. It returns a refined WBDD with less than $s$ internal nodes that perfectly classifies all examples in the labelled dataset if such a WBDD exists, and returns False otherwise.

The paper demonstrates that this algorithm runs in $\mathcal{O}((6 s^2 \delta D)^s \cdot sn))$ time, and as such improves on previous theoretical results (Ordyniak et al., 2024). Here, $s$ is the number of internal nodes of the BDD, $n$ is the number of examples in the labelled dataset, $D$ is the domain size of the variable with the largest domain, and $\delta$ is a parameter that is smaller than $D$.

The key trick to achieve this results lies in applying an algorithmic paradigm previously used for learning/constructing decision trees. The tree is built incrementally, using examples that are classified incorrectly by the tree ("witnesses") to refine the decision tree, by adding a separating hyperplane such that it classifies also that witness correctly.

The paper applies this principle to BDDs. To this end, the paper introduces a data structure called a  witness BDD (WBDD), which allows for node insertions to incrementally build the BDD. The paper discusses certain properties (consistency, perfection) of WBDDs and how they relate to BDDs that correctly classify all instances in the dataset.

The paper also derives some other computational complexity results regarding this problem, and performs a small empirical evaluation using a proof-of-concept implementation, to demonstrate the effectiveness of the algorithm against the state of the art (a SAT-based approach from (Cabodi et al., 2024)). The main observation is that `WitBDD` outperforms the SAT-based approach in terms of running time for BDDs with an $s$ of 4 or smaller.

**Claims And Evidence:**

I am inclined to believe all the claims made in the paper. However, I get the impression that this paper leans heavily on previous publications and its appendix. I find the paper very clearly written, but also very compactly. I can imagine that it would be easier to parse if the reader would've been familiar with some of the key references (which I am not). Additionally, the appendix contains a bunch of proofs that I honestly did not read, so that I would be trusting the authors on.

**Essential References Not Discussed:**

For completeness, the paper could mention [Latour et al., 2019] as another example of using BDDs for constraint propagation. The works mentioned use DDs to model the set of solutions to a set of constraints, and present filtering algorithms for those datastructures to facilitate propagation. In contrast, [Latour et al., 2019] uses a BDD to represent a *constraint* instead of a set of solutions, and presents a propagation algorithm whose complexity depends on the size of the BDD, making it a potentially relevant application for the work here presented in the paper under review.

[Latour et al., 2019] A. L. D. Latour, B. Babaki, and S. Nijssen, ‘Stochastic Constraint Propagation for Mining Probabilistic Networks’, in _Proceedings of the Twenty-Eighth International Joint Conference on Artificial Intelligence_, Macao, China: International Joint Conferences on Artificial Intelligence Organization, Aug. 2019, pp. 1137–1145. doi: [10.24963/ijcai.2019/159](https://doi.org/10.24963/ijcai.2019/159).

**Experimental Designs Or Analyses:**

The complexity results presented in this paper involve a number of parameters. In particular: the paper states that `WitBDD`'s worst-case complexity dependency on $s$ is optimal. I would have liked to see more details on the experimentation to better understand which components of the theoretical complexity result are also reflected in the empirical evaluation.

For now, I do not know how many examples there are in the datasets used for the empirical evaluation (so I do not know $n$), and I do not know the domain sizes (so I do not know $D$). I also do not know the number of classes (doesn't matter for the complexity, but I would still be interested in knowing). All in all, I just wanted to have more information about these things, and maybe also about the theoretical complexity of the baseline.

**Methods And Evaluation Criteria:**

Yes and no. I find that the main contribution of this paper is theoretical. I *really* appreciate the efforts made to create a proof-of-principle implementation and a tiny empirical evaluation. However, I found the description of the empirical evaluation too reductive to be very useful (see below for more details), and the scope of the experiment too limited.

The introduction argues that minimising BDDs is useful because of explainability and because BDDs can be more compact than decision trees. The largest BDD in the experiments has 7 internal nodes. That doesn't seem like a lot compared to the BDDs that I encounter in daily life. I understand that there's a difference between theory and practice, but now this difference seems to be so big that the empirical evaluation seems somewhat pointless to me.

**Other Comments Or Suggestions:**

- Line 19: "widely studied" -> "widely-studied"
- Line 65: "a incorrectly" -> "an incorrectly"
- Lines 118 - 127: I found this paragraph very hard to parse, because I initially thought that there should be a single $h$ that applies to all pairs of $e_1, e_2 \in E$. Maybe indexing it as $h_{1,2}$ would've made it easier to parse for me.
- Figure 2: "The arcs which" -> "The arcs that"

**Other Strengths And Weaknesses:**

I find that the paper is well-written and mostly well thought-out. There are only a few typos, and the paper contains figures to aid understanding. The structure and signposting facilitate understanding. As mentioned above, I like the attempt at empirical evaluation. I also believe that the contribution is interesting, but mostly from an academic perspective, and less so from a practical one.

**Questions For Authors:**

Q1. Why only compare to (Cabodi et al., 2024) and not to (Ordyniak et al., 2024)?
Q2. Lines 216-217: Can the authors please clarify what they mean by "Additionally, $a_2$ is allowed to be $\bot$."?
Q3. Can the authors please address my questions about the experimental parameters as stated above, and share any insights that they have regarding how the theoretical complexity dependence on different parameters is translated into an empirical one?

**Relation To Broader Scientific Literature:**

I find that the paper positions it very well in terms of the existing literature, as far as I can tell, based on my limited familiarity with the literature in this field. Claims are referenced diligently and timely.

**Theoretical Claims:**

As stated above, I feel that the paper does rely a lot on the appendix for its proofs, and maybe also a bit on previous work for easier understanding of the contents. I am not necessarily against that, but it does make me feel that this paper isn't very "standalone", and might be more suited for a publication venue that allows for a higher page limit, to aid readability and trust.

---

> ### Author Rebuttal · Authors · 2025-03-31
>
> Thanks for your review!
>
> Q1: First, Ordyniak et al. (2024) did not implement their algorithm. Moreover, it has a considerably worse theoretical running time bound ($O((3δD)^{s^2}\cdot n^{O(1)}$) and some brute-force enumerative steps that cannot be avoided. Thus, our algorithm is clearly faster since it has a smaller search space with barely any additional overhead. Moreover, our algorithm is a branch-and-bound algorithm meaning that if we select a suitable object for branching the search space can be much smaller in practice.
>
> Q2: We use $a_2 = \bot$ to signify that the one-step-refinement is a leaf-insertion since a leaf-insertion does not require a second arc.
>
>
> Q3: An overview of the 35 base instances from which we sampled our 700 test instances can be found in Staus et al., 2024 (https://doi.org/10.48550/arXiv.2412.11954, Table 3 in the Appendix). We will provide a full version of our paper on arxiv where we will include a similar overview.
> The sampled instances contain between 7 and 271 examples with a median of 31. The largest domain size D is between 2 and 145 with a median of 25 and the number of dimensions d is between 3 and 228 with a median of 21. Also, all instances have exactly two classes.
>
> Our experimental results stratified for individual parameters show large variance which makes it difficult to deduce strong statements on the empirical dependence on the various parameters besides the exponential dependence on $s$ that is evident from the theoretical running time bound. There are many instances with small values for parameters like n, delta, or the number of cuts where our algorithm is faster than the SAT algorithm but there are also some instances with similarly small values for these parameters where we are slower. At first glance, it seems that our algorithm exhibits at most a weak running time dependence on n, delta, and the number of cuts whereas the SAT running times seem to be more dependent on these parameters. To illustrate this we will add more plots for these parameters similar to Figure 3 to the full version of the paper on arxiv.
>
> Appendix A is indeed empty; before submitting we forgot to remove it.

---

### Decision · Program_Chairs · 2025-05-01

**Decision:**

Accept (poster)

**Comment:**

Solid theoretical contribution for generating classification BDDs of bounded size. It has been appreciated by the reviewers, while it remains unclear the breadth of applicability of the ideas. The reviews may give more detailed suggestions on how to improve the manuscript. All in all, impact is indeed uncertain but the work is considered enough (by most) for presentation at the conference.